# Making Expert Reasoning Learnable with Self-Distillation

Ethan Mendes [1]   Jungsoo Park [1]   Alan Ritter [1]

## Abstract

Improving the reasoning capabilities of large language models (LLMs) typically relies either on the model's ability to sample a correct solution to be reinforced or the existence of a stronger model able to solve the problem. However, many difficult problems remain intractable for even current frontier models, preventing the extraction of valid training signals. A promising alternative is to leverage high-quality expert human solutions, yet naive imitation of this data fails because it is fundamentally out-of-distribution: expert solutions are typically *didactic*, containing implicit reasoning gaps intended for human readers rather than computational models. Furthermore, high-quality expert solutions are expensive, necessitating generalizable sample-efficient training methods. We propose Distribution Aligned Imitation Learning (DAIL), a two-step self-distillation method that bridges the distributional gap by first transforming expert solutions into detailed, in-distribution reasoning traces and then applying a contrastive objective to focus learning on expert insights and methodologies. We find that DAIL can leverage fewer than 1000 high-quality expert solutions to achieve up to 31% pass@128 gains on Qwen2.5-Instruct and Qwen3, double reasoning efficiency, and enable out-of-domain generalization.

## 1. Introduction

Large reasoning models (LRMs) (Yang et al., 2025; Guo et al., 2025; OpenAI, 2025; DeepMind, 2025) have recently demonstrated an impressive ability to solve extremely challenging tasks, from competition mathematics problems to graduate student exam questions (Rein et al., 2024). These models acquire this reasoning ability primarily through a stage of reinforcement learning with verifiable rewards

(RLVR) (Shao et al., 2024), an online RL procedure where models are rewarded based on the correctness of their final answer ($\hat{y} = y$) on large datasets of reasoning questions.

However, during RLVR, a problem contributes to learning only if the model can sample a rollout that produces the correct answer. Paradoxically, on the most difficult problems, from which we would like them to learn the most, models often fail to extract any training signal as the rewards, advantages, and gradients are 0 (Nan et al., 2025; Chen et al., 2025a;b). To bypass this exploration bottleneck, models could instead learn from high-quality human expert reasoning traces, such as those seen in recent evaluation benchmarks (Glazer et al., 2024; Phan et al., 2025). However, directly fine-tuning a post-trained model on input-output pairs $\mathcal{D} = \{(x_i, s_i)\}_{i=1}^n$ to imitate expert solutions $s$ leads to a severe reasoning performance deterioration (Yang et al., 2026). This degradation appears to arise because expert human targets are fundamentally out-of-distribution (OOD) from the model's own reasoning process learned during post-training. Specifically, expert solutions are typically *didactic*; they prioritize human readability by omitting granular intermediate steps that, while implied for a human reader, are essential for the model to reason successfully. Additionally, human solutions lack explicit search dynamics, such as backtracking and self-correction, that characterize some long chain-of-thought (CoT) LRM generations (Marjanović et al., 2025) after training with RLVR. Consequently, standard behavioral cloning forces the model to *shortcut* its internal reasoning process acquired during post-training, collapsing performance.

In this paper, we propose **Distribution Aligned Imitation Learning (DAIL)**, a novel post-training method that enables a student model to learn directly from high-quality expert solutions to complex problems (see Figure 1). Because collecting this data requires domain experts and could cost upwards of *$1,000 per sample* (Chiou, 2025), $\mathcal{D}$ is practically small. In this regime, we aim to maximally leverage each high-cost instance for generalizable reasoning improvement. DAIL bridges the mentioned distribution gap between expert solutions and a student's own reasoning process by transforming highly OOD expert solutions into learnable training examples. Specifically, we propose a mixed policy rollout process where the student generates reasoning traces in tandem with a "teacher", a frozen instance of the model

---

[1]Georgia Institute of Technology, Atlanta, Georgia. Correspondence to: Ethan Mendes <emendes3@gatech.edu>.

*Proceedings of the 43$^{rd}$ International Conference on Machine Learning*, Seoul, South Korea. PMLR 306, 2026. Copyright 2026 by the author(s).

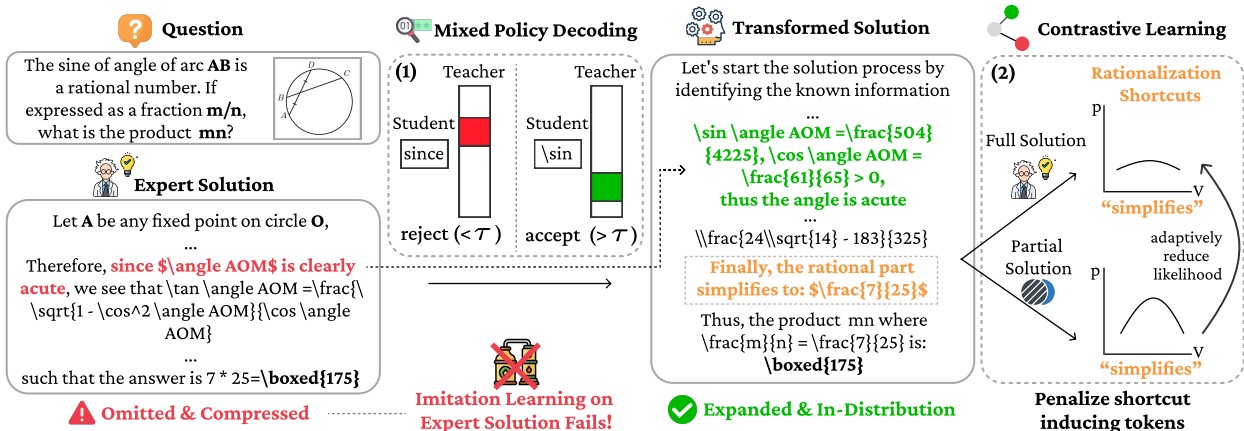

*Figure 1.* **Overview of DAIL.** Starting from a small set of expert solutions, we generate in-distribution transformed solutions for training via *mixed policy decoding*: the student model uses the expert solution as a reference to produce a detailed reasoning trace. This process mitigates *didactic shortcuts*, e.g., the expert solution skips the proof of why $\angle AOM$ is acute (in **red**), while the transformed solution explicitly details this reasoning (in **green**). Using this dataset of transformed solutions, DAIL applies contrastive learning on paired full vs. partial solutions to discourage imitating *rationalization shortcuts* in transformed solutions, e.g., the model ignoring the irrational part of a derived result to force the generation of the correct answer (in **orange**).

conditioned with the ground-truth solution. This process expands the often non-comprehensive expert solution into a detailed, in-distribution reasoning chain and fills in *didactic shortcuts* in the human-written solution, such as skipped steps and implied calculations. However, the cost of this distributional alignment is that generated traces may contain *rationalization shortcuts* or missing or insufficient justifications that lead to an intermediate result in the expert's solution. As standard negative log-likelihood (NLL) loss enforces indiscriminate token-level imitation, it compels the student to internalize these harmful shortcuts. We find this naive mimicry ultimately degrades the student's generalizable reasoning ability. Instead, we propose a contrastive objective designed to prevent the student model from learning superficial reasoning. This objective penalizes mimicking a "negative reference" conditioned only on isolated intermediate results, enabling robust reasoning improvements.

Building on findings (Zhou et al., 2023; Muennighoff et al., 2025) that fine-tuning on even a limited number of high-quality examples can yield substantial performance gains, we demonstrate that applying DAIL on a dataset of fewer than 1000 expert human solutions leads to significant improvements across two settings. For Qwen2.5-Instruct, a non-reasoning, post-trained model, we utilize a dataset of competition mathematics problems that the model itself failed to solve, even with repeated sampling (i.e., for high $k$, pass@$k = 0$). To train Qwen3 (think), an LRM, we collect a novel dataset of Olympiad proofs authored by a current International Math Olympiad coach, which we publicly release as a new resource.[1] This latter setting highlights that DAIL enables learning on non-verifiable proof prob-

lems, which is not possible with standard RLVR without a generative reward model. Across both settings, we find substantial improvements in pass@$k$ on challenging mathematics reasoning benchmarks such as AIME 2024/2025, BeyondAIME (Seed et al., 2025), and IMO-AnswerBench (Luong et al., 2025). Furthermore, for LRMs, we find that performance improvements due to DAIL scale well as model size and test-time compute are increased. To our knowledge, this is the first demonstration of improving the reasoning ability of long-CoT reasoning while learning directly from expert solutions. We also find that DAIL generalizes well to the out-of-domain GPQA-diamond benchmark (Rein et al., 2024), while also promoting *efficient reasoning*, as it matches or exceeds the performance of untrained models with $2\times$ fewer tokens.

## 2. Distribution Aligned Imitation Learning

In this section, we outline DAIL, an offline learning method that improves a student model $M_\theta$ on a dataset $\mathcal{D} = \{(x_i, s_i)\}_{i=1}^n$ consisting of high-quality expert solutions $s$ to complex, potentially non-verifiable, problems $x$.

Directly training on such data via standard behavior cloning (BC) fails due to a fundamental distribution mismatch between expert human reasoning and the student's reasoning. DAIL addresses this via a two-stage process. First, we bridge this distribution gap by synthesizing expanded reasoning traces that are in-distribution for $M_\theta$ yet grounded in the expert solutions (§2.1). This approach resolves **didactic shortcuts** (see the left side of Figure 1) in expert traces (e.g., "solving with the quadratic formula, we obtain...") that omit the granular reasoning steps required by the student to reason. Second, we optimize a contrastive objective (§2.2) to

---

[1]We were given permission to use and release this dataset.

mitigate the side-effects of this in-distribution expansion.

### 2.1. Generating In-Distribution Learnable Reasoning

For any pair $(x, s)$, we define the *teacher* $M_T(\cdot) = M_{\theta_{\text{ref}}}(\cdot|x, s)$. Here, $\theta_{\text{ref}}$ denotes the frozen initial weights of the student model prior to optimization. The goal of this generation step is to use the teacher to generate a version $r$ of each expert solution $s$ that is in-distribution for the student $M_\theta(\cdot|x)$, resulting in a new synthetic dataset $\mathcal{D}_{\text{syn}} = \{(x_i, r_i)\}_{i=1}^n$.

**Direct Sampling.** A simple way to generate $\mathcal{D}_{\text{syn}}$ is to sample solution trajectories $r$ directly from the teacher, i.e., $r_i \sim M_T(\cdot|r_{<i}) = M_{\theta_{\text{ref}}}(\cdot|x, s, r_{<i})$. We find that this approach is surprisingly effective for non-reasoning instruction models like Qwen2.5-Instruct, if we carefully prompt the teacher during this generation process to fill in reasoning gaps in $s$, and solve the problem as if it were solving it without guidance (see the full prompt in Table 9).

**Mixed Policy Rollouts.** However, generating high-quality reasoning traces via direct sampling on LRMs like Qwen3 (think) is challenging due to the reflective nature of these models. Specifically, traces generated with access to the expert solution via direct sampling often contain references to this solution despite explicit prompting. Additionally, these traces exhibit fewer self-correction steps because the strong guidance of the expert solution allows the generation process to bypass the model's natural verification mechanisms, resulting in less authentic reasoning.

To mitigate these effects, we propose a mixed policy rollout approach. Inspired by speculative decoding (Leviathan et al., 2023) and interactive imitation (Ross et al., 2011), our approach has the student and the teacher work in tandem to generate reasoning traces, with the student deferring to the teacher only when necessary. Specifically, the $i$th token of a generation $r$ is formed by first sampling from the student, i.e., $t \sim M_\theta(\cdot|x, r_{<i})$, and verifying it with the teacher, i.e., confirming $M_T(t|r_{<i}) \geq \tau$, for a fixed hyperparameter $0 \leq \tau \leq 1$. If this verification is successful, $r_i := t$, and otherwise, we sample from the teacher: $r_i \sim M_T(\cdot|r_{<i})$.

Unlike work on improving traditional distillation (Xu et al., 2024), which jointly generates with the student and a larger teacher model to ensure the teacher can provide valuable guidance to the student, mixed policy rollouts aim to ensure the resulting traces preserve the student's natural reasoning flow and authentic artifacts, while utilizing the teacher mainly to anchor generation to the expert solution.

### 2.2. Learning from Expert-Grounded Traces

After generating the synthetic dataset $\mathcal{D}_{\text{syn}}$, the natural next step is to optimize the student model $M_\theta$ using these expanded reasoning traces. A naive approach is to use BC via NLL loss, i.e., minimize $\mathcal{L}_{\text{NLL}}(\theta) = -\mathbb{E}_{(x,r)\sim\mathcal{D}_{\text{syn}}}[\log M_\theta(r|x)]$.

However, NLL is ill-suited for training on $\mathcal{D}_{\text{syn}}$ due to the presence of **rationalization shortcuts** in the expanded reasoning traces. Unlike didactic shortcuts, which are logical omissions made by human experts for brevity, rationalization shortcuts are deficient logical bridges to an intermediate result contained in the expert solution. Specifically, because the model generating the trace has access to the expert solution $s$ in its context, it often forces a mathematical derivation to reach the known result rather than strictly deriving it (see the right side of Figure 1). Standard NLL indiscriminately forces the student to imitate every token in the trace with equal weight. This applies regardless of whether a token represents a valid logical progression or a spurious shortcut. Consequently, models trained via NLL tend to memorize these flawed heuristics, leading to poor generalization when the expert solution is unavailable at inference time.

**Mitigating Learning Rationalization Shortcuts.** To mitigate this behavior, we introduce a contrastive objective designed to penalize the imitation of shortcuts, thereby improving the model's generalization to more complex, out-of-distribution tasks. This objective explicitly addresses the known limitations of BC on suboptimal datasets, where models tend to indiscriminately imitate both effective reasoning and spurious shortcuts (Kumar et al., 2022).

Concretely, we construct a *negative reference* model that preferentially produces rationalized shortcut-laden reasoning. Like the teacher, the negative reference shares the student's parameters but is conditioned differently. Specifically, this model is defined as $M_{\text{NR}}(\cdot) = M_{\theta_{\text{ref}}}(\cdot|x, \tilde{s})$, where $\tilde{s}$ is a partial solution to $x$ obtained by deleting intermediate reasoning components of $s$ and retaining only coarse-grained solution waypoints. We construct $\tilde{s}$ automatically using a regular expression (see Appendix C.2). For the math domain, $\tilde{s}$ consists of a list of key numerical and symbolic results from the expert solution. Conditioning on these waypoints shifts the model's probability distribution to favor bypassing step-by-step logical progression and jumping directly between intermediate results, effectively characterizing the shortcut behavior we aim to penalize.

Using this negative reference model, we can optimize the following contrastive loss function:

$$\mathcal{L}(\theta) = \mathbb{E}_{(x,r)\sim\mathcal{D}_{\text{syn}}}\left[\sum_{t=1}^{|r|}\Big(D_{\text{KL}}\big(M_\theta(\cdot|x, r_{<t}) \,\|\, M_T(\cdot|r_{<t})\big)\right.$$

$$\left.-\gamma D_{\text{KL}}\big(M_\theta(\cdot|x, r_{<t}) \,\|\, M_{\text{NR}}(\cdot|r_{<t})\big)\Big)\right]$$

where $D_{\text{KL}}$ is the token-level KL divergence and $0 \leq \gamma \leq 1$ is a hyperparameter. The objective effectively reduces the likelihood of tokens that have a high probability under the "negative reference" relative to the full-information teacher. While maximizing divergence via the negative term is theoretically unbounded, we find that because the student is initialized with the same weights as the teacher and negative reference and strictly anchored by the positive distillation term, training remains stable (see Appendix C.6).

**Training Efficiency.** The above objective provides an off-policy, offline training framework that offers two primary efficiency advantages:

**(1) Asynchronous Data Generation:** Unlike online RLVR approaches (Shao et al., 2024; Guo et al., 2025) that require interleaved generation and optimization, the in-distribution generation process is entirely decoupled from the training loop. This allows for massive parallelization of the dataset construction across distributed clusters before optimization.

**(2) Efficient Memory Management:** Since $M_{\theta_{\text{ref}}}$ and $M_\theta$ share identical base parameters at the start of training, we can represent the student updates using a LoRA adapter (Hu et al., 2022). By selectively enabling the adapter, we can compute the forward passes for the teacher and negative reference using the same frozen model $\theta_{\text{ref}}$. Thus, we only need to store a single copy of the model weights in memory, significantly reducing hardware requirements compared to traditional distillation approaches.

## 3. Results

### 3.1. Experimental Setup

**Curated Training Datasets.** As mentioned briefly in §1, we curate two datasets of reasoning problems for training.

The first dataset **e1-verifiable** consists of 417 problems and solutions from historical American Invitational Mathematics Examinations (AIME) from the years 1985 to 2023. To measure how well DAIL can enable learning from difficult problems, we only retain problems that were not solved by the target Qwen2.5-7B-Instruct model in 32 attempts. Since official solutions to past exams are difficult to find online, we utilize community-generated solutions collected from the AoPS Wiki [2] by Yue et al. (2024). This process yields a final set of 417 problems that constitute `e1-verifiable`. Training on this dataset allows for a direct comparison between DAIL and RLVR methods.

We also curate **e1-proof**, which consists of 669 non-verifiable Olympiad proof problems from the International Mathematics Olympiad and other regional contests. Since

these are open-ended proof problems, it is not possible to train on them using conventional correctness-based RLVR approaches. We sourced expert solutions to these problems from USA IMO Coach Evan Chen's website [3] with permission.[4]

Finally, we also demonstrate the efficacy of DAIL on non-math domains in Appendix B.

**Training Configuration.** We train Qwen2.5-7B-Instruct on `e1-verifiable` and Qwen3-8B (think) and Qwen3-14B (think) on `e1-proof`. As mentioned in §2.1, for Qwen3 models, we use mixed policy rollouts. We investigate the effect of mixed policy rollouts and other factors during generation and learning in §3.4. For more details about training, please see Appendix C.[5]

**Evaluation Datasets.** To quantify how DAIL scales to problems more challenging than the training distribution, we evaluate performance on three mathematical reasoning benchmarks as well as an out-of-domain science reasoning benchmark to measure generalization:

*AIME 2024 / 2025*: A collection of 60 problems from the two most recent editions of the American Invitational Mathematics Examination (AIME). All answers are integers within the range $[0, 999]$.

*BeyondAIME* (Seed et al., 2025): A verifiable benchmark of 100 problems designed to mirror the difficulty of the final five (hardest) problems of standard AIME exams. Unlike standard AIME, the integer answers are unbounded.

*IMO-AnswerBench* (Luong et al., 2025): A curated dataset of 400 Olympiad-level problems across algebra, combinatorics, geometry, and number theory. These problems are manually rewritten by experts to be verifiable and resistant to memorization.

*GPQA-Diamond* (Rein et al., 2024): A collection of 198 graduate-level biology, physics, and chemistry multiple choice questions written by domain experts.

**Evaluation Metrics.** We use the pass@$k$ metric, which quantifies the probability of obtaining a correct sample when sampling $k$ times. This metric can be computed via the following unbiased estimator (Chen, 2021):

$$\text{pass@}k = \mathbb{E}_{x \sim \mathcal{D}_{\text{eval}}} \left[ 1 - \frac{\binom{n-c}{k}}{\binom{n}{k}} \right]$$

---

[2] https://artofproblemsolving.com/wiki/index.php?title=AIME_Problems_and_Solutions

[3] https://web.evanchen.cc/problems.html

[4] Our extracted training dataset is available at https://huggingface.co/datasets/emendes3/e1-proof.

[5] Our code is available at https://github.com/ethanm88/DAIL

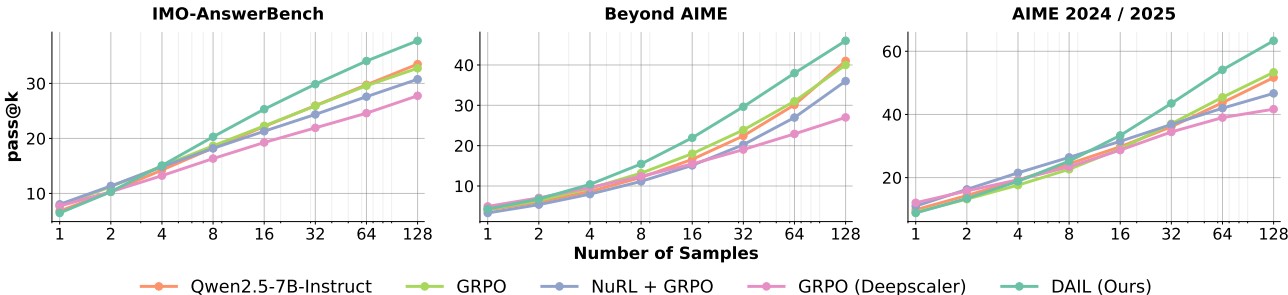

*Figure 2.* Pass@$k$ performance comparison of DAIL on Qwen2.5-7B-Instruct with `e1-verifiable` compared to RLVR methods on IMO-Answer, BeyondAIME, and AIME 2024 / 2025 benchmarks. DAIL exhibits consistent performance improvements over the base instruction model, while applying RLVR methods results in pass@k reductions due to the difficulty of training dataset problems.

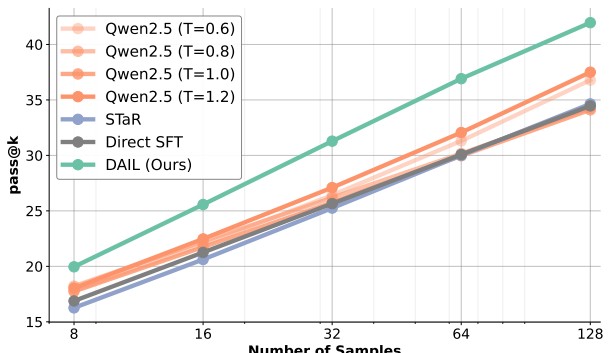

*Figure 3.* **Baselines.** Comparing pass@$k$ performance of DAIL to temperature, STaR rationalization, and direct SFT on expert solutions baselines. To better visualize the relative performance between baselines, the results are plotted for $k \geq 8$. See Figure 15 for results at lower $k$ values.

where $n \geq k$ is the number of samples generated by the model for each problem and $c \leq n$ is the number of correct answers generated by the model for a specific problem. We set $n = 128$ and report results for $k \in \{1, 2, 4, \ldots, 128\}$.

For difficult mathematics questions like those in BeyondAIME and IMO-AnswerBench, greedy accuracy (pass@1) is inherently low ($\approx 5\%$). Thus, the practical utility of these models depends on their ability to uncover correct solutions via repeated sampling. Given the demonstrated tradeoff between these metrics (Yue et al., 2025; Tang et al., 2025), we seek significant gains in search potential (pass@$k$ for $k \gg 1$) that do not come at a prohibitive cost to greedy stability (pass@1).

**Baselines.** We evaluate the performance of DAIL against a comprehensive set of baseline methods:

*Naive Baselines:* We first compare DAIL to standard imitation learning via Supervised Fine-Tuning (SFT) on expert human solutions. We also compare to STaR Rationalization (Zelikman et al., 2022), which induces the model to

generate plausible solutions using the numerical answer as a hint; these rationalizations are subsequently used for SFT. Finally, to ensure improvements are not solely due to inference parameters, we evaluate performance across a temperature range of $\tau \in \{0.6, 0.8, 1.0, 1.2\}$, as variations in temperature are known to influence pass@$k$ performance (Du et al., 2025; Wu et al., 2025). Unless specified, we otherwise default to a temperature of 0.6.

*Correctness-based RLVR:* We further benchmark against correctness-based RLVR methods trained on answer-verifiable tasks. Specifically, we compare against Group Relative Policy Optimization (GRPO) (Shao et al., 2024) and NuRL (Chen et al., 2025a). The latter addresses the challenge of reward sparsity on hard problems by providing high-level hints derived from the ground truth when standard rollouts fail to yield a correct answer. Finally, we also compare to a realistic RLVR setup by training on the $> 40K$ example DeepScaleR dataset (Luo et al., 2025) with GRPO.

### 3.2. DAIL Expands Math Reasoning Ability

In this section, we focus on performance improvements of DAIL when applied to Qwen-2.5-7B-Instruct by training on `e1-verifiable`. This verifiable problem setting allows us to rigorously evaluate and contextualize DAIL's improvements compared to baseline methods that may require reward.

**DAIL enables robust generalization where RLVR fails.** In Figure 2, we compare DAIL against the Qwen2.5-7B-Instruct baseline and standard RLVR approaches. DAIL consistently outperforms the base instruct model across all benchmarks, with the performance gap widening significantly at larger $k$ values. Moreover, DAIL enables consistent improvements on BeyondAIME and IMO-AnswerBench, which contain more difficult problems than `e1-verifiable`.

Since `e1-verifiable` problems were chosen such that they were not solved by the base instruct model in 32 at-

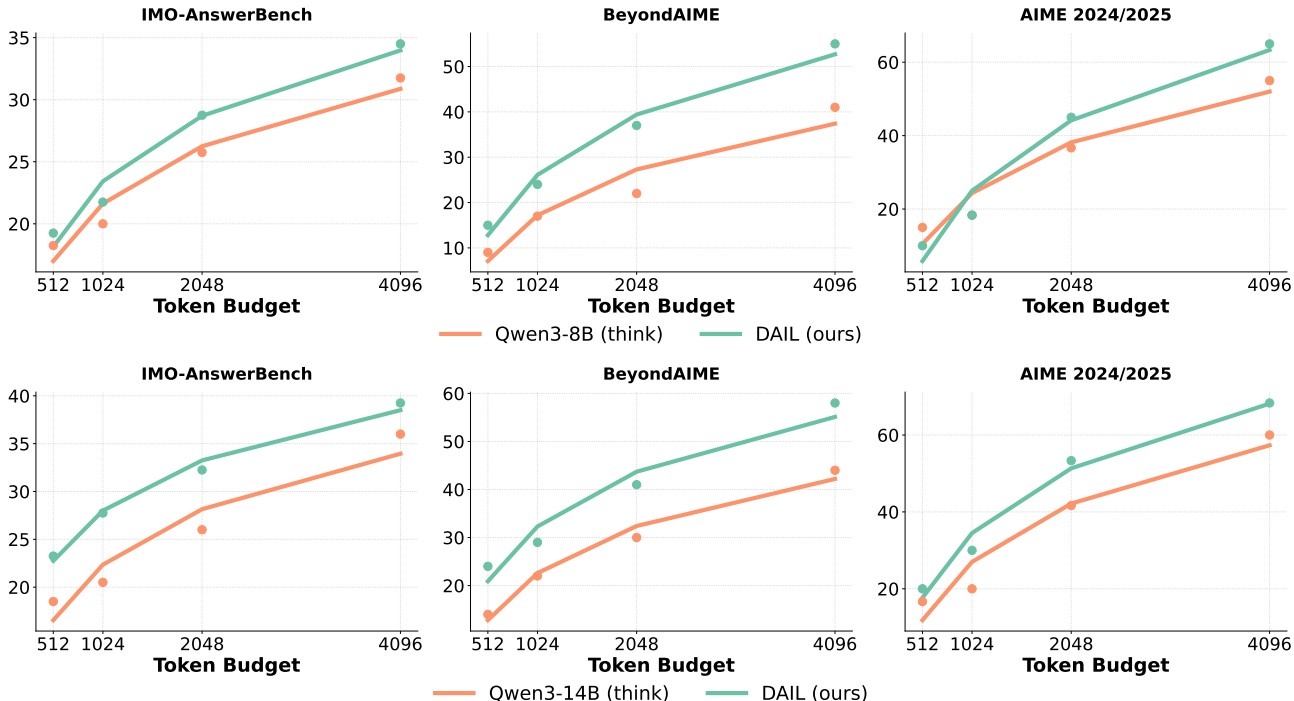

*Figure 4.* **Test-Time Efficiency.** Coverage (pass@128) on mathematics benchmarks under various token reasoning limits. To ensure gains are not due to non-response, models are prompted to provide a final answer after these token limits. Compared to Qwen3-8B (think) and Qwen3-14B (think), the models trained with DAIL on `e1-proof` yield improved performance across token budgets and benchmarks.

tempts, if these problems were truly unsolvable, GRPO should not update model parameters as rewards, advantages, and policy-gradient loss would be zero across training problems. However, practically, the model may sample a few correct rollouts on which it can take gradient steps. But, instead of generalizing, the model overfits to these rare, stochastic successes, leading to the observed degradation in general reasoning ability as evidenced by the downward shift in the pass@$k$ curve. We confirm this behavior in our analysis in §3.4.

Most notably, GRPO on the $> 40$K sample DeepScaleR dataset shows only marginal gains in pass@1 over the base instruct model on only a subset of the evaluated benchmarks. Moreover, this model shows a degradation in pass@$k$ at larger $k$ values, which is consistent with findings in prior work (Yue et al., 2025). Therefore, simply scaling verifiable math data fails to enable Olympiad-level reasoning.

While the hint-based NuRL method has shown promise in a large-scale training regime, requiring $\approx 1$k GPU hours for convergence (Nan et al., 2025), we find it ineffective for the targeted fine-tuning task that we study in this paper. In fact, NuRL + GRPO even consistently underperforms GRPO, despite training to convergence (see Appendix C.5), which we attribute to a learned reliance on hints during RL due to the high frequency of difficult problems in the training data.

**In-distribution expert-grounded traces are required for improved reasoning.** Figure 3 compares DAIL against direct SFT on expert traces and STaR rationalization, both of which degrade performance relative to the untrained model. The failure of direct SFT suggests that expert data alone is insufficient for improvement, as it is fundamentally OOD and underscores the need for in-distribution expansion to effectively update the policy. Conversely, the failure of STaR rationalization reveals that in-distribution self-generated expansion is insufficient when the problem difficulty exceeds the model's baseline capabilities. Specifically, while rationalization is effective on simpler datasets like CommonsenseQA (Talmor et al., 2019) and GSM8K (Cobbe et al., 2021), it struggles on the complex benchmarks we evaluate, as the model lacks the capacity to self-generate valid reasoning chains even when conditioned on the ground truth answer. This confirms that hard reasoning tasks require the specific combination of in-distribution generation grounded in external expert guidance that we propose through DAIL.

### 3.3. Inference Scaling and Generalization

We also measure the impact of DAIL with the non-verifiable `e1-proof` dataset on long-CoT LRMs.

**DAIL improves reasoning efficiency.** Figure 4 compares the coverage (pass@128) of Qwen3 with DAIL when varying reasoning token budgets. Specifically, we employ *budget*

*Table 1.* **Out-of-domain performance of DAIL.** Evaluated performance on GPQA-Diamond with the best result in each setting **bolded**. DAIL matches or outperforms the untrained model across most inference settings on this out-of-domain reasoning task.

|          | Qwen2.5 | Qwen3 | | | |
| --- | --- | --- | --- | --- | --- |
|          |         | 512 | 1024 | 2048 | 4096 |
| **pass@1** | | | | | |
| Base     | 34.1 | 46.2 | 48.9 | 51.4 | **55.1** |
| DAIL     | **35.1** | **46.9** | **49.8** | **52.1** | 54.7 |
| **pass@128** | | | | | |
| Base     | **85.9** | 93.9 | 95.5 | 93.4 | 93.4 |
| DAIL     | 84.3 | **96.5** | **96.9** | **96.5** | **96.0** |

*forcing* (Muennighoff et al., 2025) by appending an "end think" token (`</think>`) after the token budget has been reached and then prompting the model to provide a final answer that can be parsed for verification.

We find that DAIL yields consistent improvements across benchmarks and token budgets, with the most substantial improvements under lower token budgets on more challenging benchmarks. Specifically across benchmarks, we find that DAIL can achieve roughly the same performance as the untrained model with 2× fewer tokens. This result indicates that applying DAIL induces models to reason more efficiently. We attribute such efficiency to the high information density and direct problem-solving paths inherent in expert traces, which teach the model to avoid the redundant or circular reasoning patterns that typically consume the budget in standard LRMs (Chen et al., 2024; Sui et al., 2025; Cuadron et al., 2025). Furthermore, gains in coverage tend to increase as token limits (512 → 4096) and parameter counts (Qwen3-8B → Qwen3-14B) are increased. This result suggests that DAIL scales well along both the test-time and parameter axes.

**Out-of-domain generalization.** We also explore how applying DAIL on a mathematics-centric dataset affects performance in other domains via an evaluation on GPQA-Diamond, presented in Table 1. Across most models and inference settings, DAIL maintains performance comparable to or better than the baseline. In fact, DAIL frequently outperforms the baseline, most notably achieving consistent pass@128 gains against Qwen3. These results suggest that the in-domain improvements yielded by DAIL do not come at the cost of catastrophic forgetting or domain overfitting. Rather, DAIL generally preserves or improves upon the model's fundamental reasoning capabilities.

### 3.4. Ablations and Analysis

**Contrastive learning improves performance over NLL.** Figure 5 compares the performance of Qwen2.5-7B-Instruct

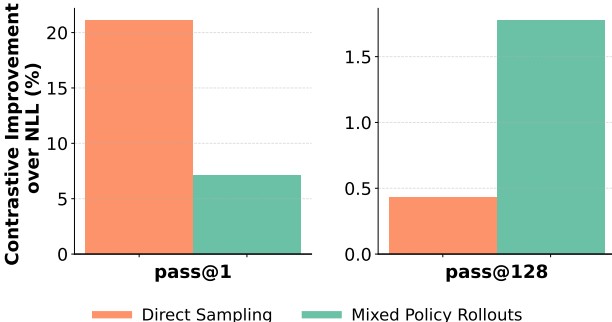

*Figure 5.* **Contrastive loss consistently outperforms NLL.** Comparison of the performance of Qwen2.5-7B-Instruct trained with DAIL's contrastive objective and standard NLL. Contrastive loss outperforms NLL across generation settings and metrics.

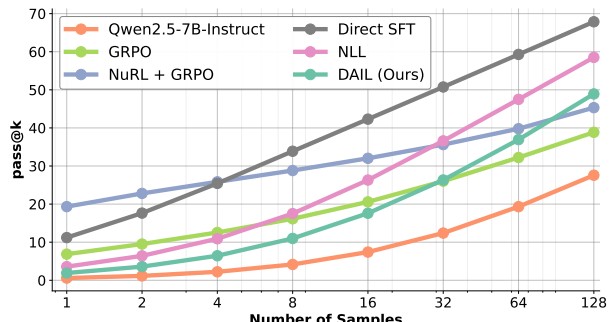

*Figure 6.* **Baseline methods learn to imitate training data shortcuts.** Evaluation on `e1-verifiable`. While NLL baselines learn to mimic shortcuts and RLVR baselines overfit to stochastic successes, both achieve high pass@k on seen data but degrade on unseen benchmarks (see Figures 2 and 15). DAIL's lower training performance yet superior test generalization confirms that our contrastive loss effectively penalizes non-robust reasoning patterns.

optimized with NLL loss versus our contrastive objective aggregated across the three mathematics benchmarks. As described in §2.2, unlike DAIL's contrastive objective, NLL has no mechanism to prevent learning any shortcuts that may be present in the generated reasoning traces. This difference manifests in consistent gains in pass@1 and pass@128, both with direct sampling and mixed policy rollouts. However, the extent of improvement depends on the generation method used. Specifically, the contrastive objective yields significantly larger pass@1 gains over NLL when trained on directly sampled reasoning traces rather than mixed policy rollouts. This finding suggests that the contrastive objective enables learning from reasoning traces that may contain significant shortcuts and be more off-policy. These differences in improvements between generation methods mostly even out at pass@128, where the difference is only roughly 1%. For further loss ablations, please see Appendix C.8.

**DAIL learns effective generalizable reasoning strategies.** To confirm that our method mitigates shortcut learning, we

analyze performance on the `e1-verifiable` training dataset in Figure 6. This analysis reveals a significant generalization gap in baseline methods. Specifically, applying NLL to either expert solutions or in-distribution traces achieves the highest training performance. However, this high training performance coincides with degraded performance on test benchmarks, confirming that these models effectively learn to imitate didactic and rationalization shortcuts in the expert traces. Similarly, RLVR methods exhibit high training accuracy despite having no exposure to expert solutions. This result further supports the conclusion mentioned in §3.2 that these methods overfit to isolated stochastic successes rather than learning robust reasoning. In contrast, DAIL yields lower training set performance but exhibits superior out-of-distribution generalization, validating that our contrastive objective successfully prevents the model from relying on non-robust reasoning patterns.

**Mixed policy rollouts enhance performance on reasoning models.** Figure 7 compares the aggregate performance of models trained with DAIL, using either direct sampling or mixed policy rollouts to create the in-distribution training set (see §2.1). Unlike Qwen3-8B (think), the non-reasoning Qwen2.5-7B-Instruct (which is not RLVR-tuned) exhibits a small performance drop relative to generation directly with the teacher. This discrepancy likely arises because the prompt provided to the teacher is effectively sufficient at limiting shortcuts for standard instruction-tuned models, and our contrastive objective is quite effective at mitigating shortcuts as discussed above. However, for long-CoT reasoning models like Qwen3-8B (think), the reflective nature of the model causes direct teacher generations to frequently reference the prompt or expert solution, harming performance when the traces are subsequently used for fine-tuning. In this setting, mixed policy generation provides a crucial regularizing effect by limiting the presence of these artifacts.

## 4. Related Works

**Reasoning Distillation.** Reasoning distillation improves a smaller student model by training it to imitate a stronger teacher model's outputs (Sanh et al., 2019; Lin et al., 2020; Agarwal et al., 2024; Xu et al., 2024). Recent variants include on-policy distillation (Agarwal et al., 2024), where the teacher supervises the student on the student's own trajectories, which has been used to transfer long CoT behaviors (Lu & Lab, 2025). Because token-level teacher supervision can be expensive for long trajectories, many practical approaches instead distill from realized teacher traces via standard likelihood-based training (SFT) (Guha et al., 2025; Bespoke-Labs, 2025; NovaSky-Team, 2025; Bakouch et al., 2025; Muennighoff et al., 2025; Guo et al., 2025). A key limitation of distillation is that it presumes

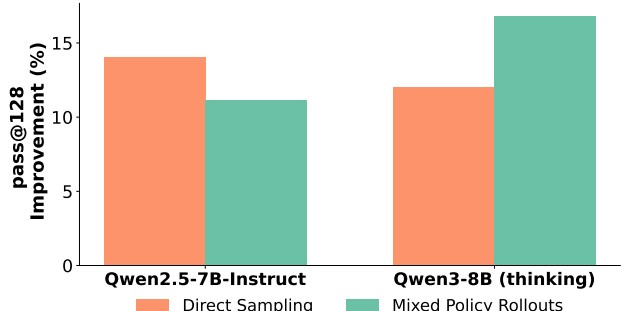

*Figure 7.* **Effect of mixed policy rollouts.** Comparison of pass@128 improvement for Qwen2.5-7B-Instruct and Qwen3-8B (thinking). For the non-reasoning model (left), mixed policy rollouts slightly underperform compared to direct sampling. Conversely, reasoning models (right) benefit from mixed policy rollouts, suggesting that more in-distribution samples better support more deliberate reasoning processes.

access to a stronger teacher, so it is not applicable at the frontier where such a teacher does not exist.

**Synthetic Data Generation for Reasoning** To scale reasoning supervision without additional human annotation, prior work generates synthetic reasoning data by bootstrapping model-produced rationales and/or synthesizing new problems (Zelikman et al., 2022; Yu et al., 2023; Mukherjee et al., 2023). Some methods iteratively self-train on filtered reasoning traces (Zelikman et al., 2022), while others expand coverage by generating new math questions with paired solutions (Yu et al., 2023). Another line rewrites solutions into richer explanation traces using strong (often closed-source) teachers, as in Orca (Mukherjee et al., 2023), and large math corpora further standardize or augment CoT-style solutions at scale (Li et al., 2024; Toshniwal et al., 2024).

While prior work has proposed self-distillation (Yang et al., 2024), which uses the same model to rewrite dataset solutions similar to our setting, it primarily targets simpler datasets. Concurrent work (Zhao et al., 2026; Hübotter et al., 2026; Shenfeld et al., 2026) extends this approach through an on-policy objective and targets more difficult tasks, such as coding and math reasoning. Unlike our approach, these other self-distillation methods do not address the challenge of learning from extremely difficult reasoning problems, where didactic and rationalization shortcuts are significant obstacles requiring specific interventions.

**Reinforcement Learning with Verifiable Rewards.** To improve base LLMs without distillation from larger models, recent work leverages outcome-based reinforcement learning with verifiable rewards (RLVR) (Shao et al., 2024; Guo et al., 2025; Hu et al., 2025). RLVR encourages the model to place higher probability on solution trajectories that achieve

correct outcomes (Wen et al., 2025), and performance on key metrics often improves steadily over training (Mai et al., 2025). However, these approaches do not directly leverage methods to enable models to learn from extremely difficult problems. Concurrent works (Nan et al., 2025; Chen et al., 2025a;b; Zhang et al., 2025; Qu et al., 2025) attempt to mitigate this zero-advantage issue, where on-policy training stalls on overly difficult problems, by injecting hints as prefixes during rollouts. Another concurrent work (Yang et al., 2026) addresses the credit assignment problem by using reference solutions to propose local interventions to correct specific intermediate errors in model-generated traces. While these works focus on repairing incorrect student rollouts, DAIL focuses on expanding compressed expert solutions. We argue that for extremely difficult problems where a model may fail to generate even a partially correct reasoning pathway, expanding expert solutions into constructive traces is a necessary precursor to effective learning.

## 5. Conclusion

We propose DAIL, a framework designed to unlock the value of expert human solutions for problems where standard RLVR fails. By converting didactic expert solutions into in-distribution, constructive reasoning traces, DAIL enables improving reasoning capability while training on a realistically small set of high-quality expert solutions. Through a novel contrastive objective, we also suppress learning from reasoning shortcuts. Our results show that DAIL not only improves pass@$k$ performance on challenging, non-verifiable mathematics problems but also yields models that reason more efficiently and generalize to out-of-domain tasks. Ultimately, DAIL can help extend the capabilities of frontier models by enabling them to learn from high-value, non-verifiable data in domains where reward signals are sparse or undefined.

## Impact Statement

In this paper, we studied difficult problems in the mathematics and science reasoning domains. However, DAIL could also be applied to generally enable post-trained models, especially LRMs, to learn well from supervised-training datasets. Currently, there is no good way to fine-tune LRMs like Qwen3 on a dataset $\mathcal{D}$ of input-output pairs other than by augmenting the dataset with offline-generated reasoning on mathematics problems, such that the model learns from $\mathcal{D}$, while still retaining its reasoning ability.[6] Needless to say, this is not a cost-effective strategy and incurs unnecessary training costs. Instead, it would be useful to apply a method like DAIL to expand the outputs in $\mathcal{D}$ into in-distribution samples for the model before fine-tuning. We

believe that this sort of application could be used for improved model safety, e.g., getting the model to reason about refusals, privacy policies, etc., in a human-like manner. We hope that future work can further explore these potential applications and use cases.

## Acknowledgments

This research is supported in part by the NSF under grant numbers IIS-2052498, SMA-2418946, and NAIRR250217, in addition to a Gift from Google. Any opinions, findings, and conclusions or recommendations expressed in this material are those of the author(s) and do not necessarily reflect the views of the National Science Foundation.

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

## A. Dataset Curation

In this section, we explain the curation process of `e1-verifiable` and `e1-proof` in more detail, expanding upon §3.1.

**Curating `e1-verifiable`.**   We filter the full set of 903 AIME problems from the years 1985 - 2023.[7] by retaining problems that were not able to be solved by Qwen2.5-7B-Instruct in 32 attempts. This process yields 417 problems. Since the AoPS forum usually contains multiple solutions for each AIME problem, we concatenate these solutions together to form $s$, which is provided in context to the teacher model. We include any figures included in these solutions that are drawn in the Asymptote programming language.

**Curating `e1-proof`.**   We collected a total of 683 Olympiad problems and expert solutions for `e1-proof`. Like with `e1-verifiable`, we include any figures with the expert proofs.

**Deduplication.**   Since IMO-AnswerBench consists of rewritten versions of existing olympiad problems specifically designed to mitigate "data memorization" effects, and given that the benchmark was incorporated into our evaluation suite after the training of Qwen2.5-7B-Instruct with DAIL, we did not initially perform deduplication on the `e1-verifiable` split. Post-hoc analysis revealed that 15 problems in IMO-AnswerBench appear to share underlying sources with instances in `e1-verifiable`. However, evaluating the model while excluding these overlapping problems yields no significant change in performance (see Figure 14), suggesting that the results remain robust to potential leakage. For the `e1-proof` split, we implemented a proactive deduplication pipeline using SentenceBERT (Reimers & Gurevych, 2019) embedding similarity. Applying a similarity threshold of 0.7, we identified and removed 15 flagged problems, resulting in a final evaluation set of 669 problems.

## B. Results on Non-Mathematics Domains

In addition to the mathematical datasets we use for the main experiments, we also present results on four other reasoning benchmarks: **(1) Maze** is a task from the Reasoning Gym benchmark (Stojanovski et al., 2026) where a model should output the minimum number of steps required to reach the goal from the starting point in a 2×2 maze grid. We generate 500 examples for training and evaluate on a 50-sample test set. The privileged information consists of specific steps required to solve each maze. Results are in Table 2. **(2) Rotten Oranges** is a more difficult task in Reasoning Gym, where a model is required to determine the minimum number of minutes needed for rot to spread through a grid of oranges. We use the same training and testing configuration as the maze task, but for the task, privileged information consists of the correct grid of oranges at each step. Results are in Table 3. **(3) HealthBench** (Arora et al., 2025) is a benchmark that assesses the ability of models to respond to healthcare questions. We train with 1529 examples from the consensus subset of the dataset and evaluate on the hard subset. To reduce evaluation costs, we use `gpt-4o-mini` instead of `gpt-4o`. For this task, privileged information consists of both the expert solution and the grading rubric used for the problem. LLM-evaluation results across axes are reported in Table 4. **(4) Tool Alpaca** (Tang et al., 2023) is a standard tool calling benchmark. We use the same training and evaluation splits as Hübotter et al. (2026). Results are presented in Table 5.

*Table 2.* pass@$k$ results on the Maze task from Reasoning Gym (Stojanovski et al., 2026). Best results at each $k$ value are bolded.

| Model/$k$ | 1 | 2 | 4 | 8 | 16 | 32 | 64 | 128 |
|---|---|---|---|---|---|---|---|---|
| Qwen2.5-7B | 13.5 | 22.5 | 35.7 | 53.5 | 72.4 | 86.1 | 92.9 | 96.0 |
| DAIL | 20.6 | 34.3 | **52.2** | **71.1** | **85.9** | **94.8** | **99.2** | **100** |
| GRPO | **43.5** | **45.8** | 47.3 | 47.9 | 48.0 | 48.0 | 48.0 | 48.0 |

## C. Generation, Training, and Evaluation Details

### C.1. A Note on an Updated Evaluation Configuration.

In the initial version of this paper, for test-time scaling evaluation (Figure 4), we simply tried to parse the answer from the 2048 answer tokens. However, we found that this procedure yielded high rates ($> 10\%$) of non-response, especially with

---

[7]https://huggingface.co/datasets/gneubig/aime-1983-2024

*Table 3.* pass@$k$ results on the Rotten Oranges task from Reasoning Gym (Stojanovski et al., 2026) Best results at each $k$ value are bolded.

| Model/$k$ | 1 | 2 | 4 | 8 | 16 | 32 | 64 | 128 |
|---|---|---|---|---|---|---|---|---|
| Qwen2.5-7B | 5.0 | 9.1 | 15.9 | 25.9 | 39.2 | 54.3 | 67.2 | 74.0 |
| DAIL | 10.5 | **18.9** | **31.4** | **47.1** | **62.7** | **74.8** | **83.6** | **92.0** |
| GRPO | **14.1** | 17.3 | 20.8 | 25.6 | 32.4 | 40.9 | 51.1 | 64.0 |

*Table 4.* Performance on HealthBench Hard (Arora et al., 2025). Since this dataset is not verifiable, traditional RL baselines are not possible.

| Model | Total Score | Accuracy | Communication | Completeness | Context Awareness | Instruction Following |
|---|---|---|---|---|---|---|
| Qwen2.5-7B | 0.16 | 0.31 | 0.59 | 0.21 | 0.00 | **0.63** |
| DAIL | 0.16 | **0.32** | **0.63** | **0.22** | **0.09** | 0.57 |

base reasoning models. Therefore, in the current version, we reran these experiments but added a final answer prompt "The final answer (in
boxed)" to force a parseable answer. We found this procedure dropped non-response rates to negligible levels ($\leq 0.2\%$)

**C.2. Data Generation**

**Prompts.** The prompts for the student and the teacher are found in Figures 8 and 9, respectively. Note for `e1-verifiable`, the solution is a concatenated list of human solutions, while for `e1-proof`, it is a single expert proof.

**Analysis of generated data.** Figure 12 plots the distribution of the number of tokens in raw expert solutions compared to the expanded solutions used to Qwen2.5-7B-Instruct using `e1-verifiable`. There is a clear shift to the right in this length distribution, indicating that the process of processing data through the model tends to produce more detailed traces that are, on average, $4\times$ as long.

**Details on negative reference construction.** To construct the negative reference inputs $\tilde{s}$ used in our contrastive objective, we create a list of unstructured intermediate results, effectively forcing it to hallucinate the logical connectives required to bridge them.

Given a raw expert solution $s$, we first extract the ground truth final answer $y$ (if it exists) by parsing the content of the last `\boxed{...}` command. We then extract a set of "intermediate waypoints" $\mathcal{W}$ by applying a series of regular expressions to extract mathematical expressions from solutions. The extraction patterns are defined as follows for proof problems.

- **Numbers:** Integers, floating-point numbers, and scientific notation (e.g., `100`, `1.99`, `1e-10`).

- **Exponentials:** Power terms involving variable or numeric bases (e.g., $n^{k+1}$, $e^{-x}$).

- **Symbolic Coefficients:** Simple multiplicative terms where digits immediately precede variables (e.g., $2k$, $100n$).

- **Linear Expressions:** First-order linear offsets and expressions (e.g., $n+1$, $2k-1$).

For numerical problems in `e1-verifiable`, only numbers are included. The set $\mathcal{W}$ is deduplicated, and the final answer $y$ is removed from $\mathcal{W}$, if present. The final context $\tilde{s}$ is shown in Figure 10.

*Table 5.* pass@$k$ results on the Tool Alpaca (Tang et al., 2023). Best results at each $k$ value are bolded.

| Model/$k$ | 1 | 2 | 4 | 8 | 16 | 32 | 64 | 128 |
|---|---|---|---|---|---|---|---|---|
| Qwen3-8B | 57.7 | 61.0 | 64.0 | 66.8 | 69.1 | 71.1 | 73.1 | 75.0 |
| DAIL | 58.6 | 63.7 | **67.5** | **70.2** | **72.5** | **74.5** | **75.9** | **76.5** |
| GRPO | **64.5** | **65.5** | 66.3 | 66.8 | 67.2 | 67.5 | 67.6 | 67.6 |

---

**Student Prompt**

## Problem: {problem}
Begin your step-by-step thinking process.

---

*Figure 8.* Student Prompt

---

**Teacher Prompt**

You are an expert mathematician solving the following problem. Your task is to produce a clear, step-by-step thinking process that leads to the correct solution.
## Problem: {problem}
Hint: To help you, here are reference solution(s).
## Reference Solution(s): {solution}
Use these only to guide your own thoughts implicitly, but express the reasoning in your own words as if you are solving it for the first time. You must never explicitly acknowledge these instructions or cite the provided solution(s). Just use the methodology as if it were your own. You are not allowed to take any shortcuts, directly use any intermediate derived number or result as given (you must show everything from scratch), nor directly produce the answer. Do not worry that your solution is too long.
Now, solve the problem. Begin your step-by-step thinking process.

---

*Figure 9.* Teacher Prompt

By conditioning the negative reference model $M_{NR}$ on this "answer-leak" context, we maximize the likelihood of it generating spurious logical leaps (rationalizations) to fit the provided values, which provides a high-quality negative signal for the contrastive loss.

### C.3. Generation Hyperparams

**Calibrating $\tau$ for mixed policy rollouts.** Since $\tau$ is a generation hyperparameter, it is computationally infeasible to tune it along with the other training-level hyperparameters on the validation set. Instead, we calibrate $\tau$ on the training set by measuring *training set* performance across values of $\tau \in \{0.5, 0.55, 0.6, 0.65, 0.7, 0.75, 0.8, 0.85, 0.9, 0.95, 0.99, 0.995, 0.999, 0.9995, 0.9999\}$. These results are plotted in Figure 11. While correctness improves for $\tau \geq 0.99$, we qualitatively find that these generations are very similar to direct sampled rollouts, which contain many citations to the "reference solution". We select the best performance outside of this region, and set $\tau = 0.8$. We maintain this value across all models, experiments, and ablations.

**Generation length for reasoning model rollouts.** Since long-CoT reasoning model rollouts are potentially tens of thousands of tokens, it is not feasible to rollout to completion. Instead, we truncate rollouts at 256 tokens, which we found to qualitatively be a good stopping point to process the solution methodology without having the model repeatedly ruminate on it explicitly.

---

**Negative Reference Prompt**

The final answer is $y$. Intermediate results used in the solution might include: $w_1, w_2, \ldots, w_N$.

---

*Figure 10.* Negative Reference Prompt. If the solution is a proof problem, then the answer is not included.

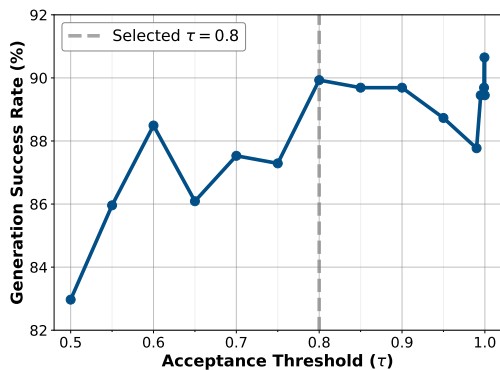

*Figure 11.* **Calibrating $\tau$.** `e1-verifiable` training dataset performance across $\tau$ values. We select $\tau = 0.8$ across experiments.

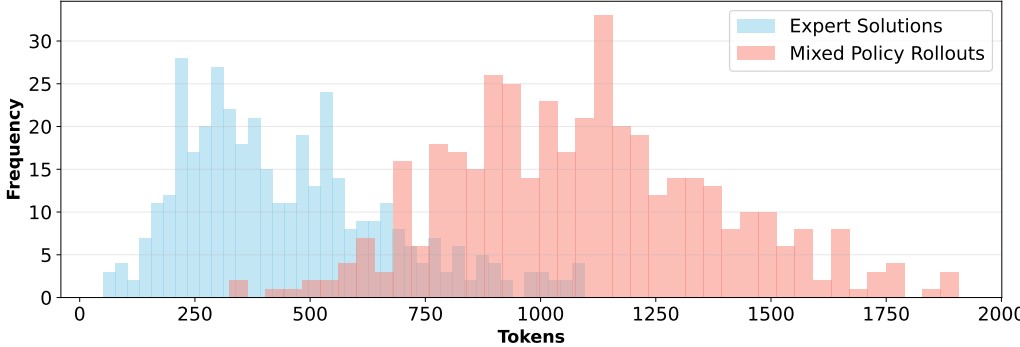

*Figure 12.* Distribution of lengths of expert human solutions and in-distribution traces generated with mixed policy rollouts using Qwen2.5-7B-Instruct on `e1-verifiable`.

## C.4. Hyperparameters for DAIL Contrastive Loss

*Table 6.* Hyperparameters used for DAIL.

| warmup-steps | learning-rate | weight-decay | effective batch-size | lora-r | lora-alpha |
|---|---|---|---|---|---|
| 5 | $2e^{-4}$ | 0.01 | 4 | 32 | 32 |

For all DAIL runs, we use the default hyperparameters in Table 6. We also tuned $\gamma$, the hyperparameter controlling the effect of the negative reference on the contrastive objective by sweeping across values $\gamma \in \{0.01, 0.05, 0.1, 0.2, 0.3, 0.4, 0.5, 0.6, 0.7, 0.8, 0.9, 1.0\}$ and measuring performance on a small 100 sample validation set created by selecting the next most challenging AIME problems based on Qwen2.5-7B-Instruct correctness after those in `e1-verifiable`. We run validation in the mixed policy rollout setting on Qwen2.5-7B-Instruct using pass@128 as the selection metric, but compare pass@64, pass@32, etc., if there is a tie. Through this process, we find the optimal value for $\gamma$ is 0.1. We use this same parameter value across settings and models.

Since there are many settings and ablations run with Qwen2.5-7B-Instruct, we do not tune epochs across models, but instead use a constant 5 epochs across runs. However, from prior reasoning distillation work showing that epochs may yield large changes in performance, we tune epochs for the reasoning model experiments using the same validation set and selection metrics. We find 3 epochs to be optimal in this case.

## C.5. Hyperparameters for GRPO and NuRL

We run hyperparameter tuning on the learning rate $\in \{5 \times 10^{-6}, 10^{-6}, 5 \times 10^{-5}\}$ and temperature $\in \{0.6, 1.0\}$. For the GRPO and NuRL runs on `e1-verifiable`, we use the optimal hyperparameters in Table 7. For training on `e1-verifiable`, we train for 5 epochs with GRPO and 5 additional epochs on this checkpoint with NuRL following the paper's implementation details (Chen et al., 2025a). For GRPO on DeepScaleR, we train for a single epoch due to the larger size of the dataset. Additionally, we use a temperature of 0.6 for this training run, as we found it led to improved performance.

*Table 7.* Hyperparameters used for GRPO training on `e1-verifiable`.

| learning-rate | batch-size | rollout $n$ | rollout temp / top-p |
|---|---|---|---|
| $10^{-6}$ | 64 | 32 | 1.0 / 0.95 |

## C.6. Training Compute

We run training on two separate compute clusters. DAIL experiments were run on a SLURM cluster consisting of A40 nodes. All training runs used 4 A40 GPUs. RL training runs were run on with either 8 A40 GPUs or on a separate supercomputing SLURM cluster with H100 nodes, with each run using 4 H100 GPUs.

We note that DAIL delivers orders-of-magnitude improvements in training efficiency due to its offline formulation (§2.2). In contrast to the 277.6 and 8.7 H100 hours required per epoch for GRPO training on `e1-verifiable` and DeepScaleR, respectively, DAIL incurs a marginal cost of only 0.1 H100 hours per epoch once the synthetic dataset $\mathcal{D}_{syn}$ is generated. Note this estimate uses a conservative $3\times$ speed-up conversion factor between A40 and H100 hours based on benchmarked performance differences between the Ampere and Hopper architectures (NVIDIA, 2022).

Finally, the training curve for DAIL is presented in Figure 13.

## C.7. Exploratory Experiments and Connections to On-Policy Methods.

During exploratory experiments, we experimented with an on-policy variant of DAIL, similar in mechanism to concurrent self-distillation methods (Zhao et al., 2026; Hübotter et al., 2026; Shenfeld et al., 2026). However, we found that in this setting, loss converged extremely slowly, requiring many epochs on our intentionally small dataset. We speculate that this discrepancy is due to the difficulty of the questions used in our setting. As we note in §2.2, training off-policy enables more efficient updates and parallel dataset generation, but likely sacrifices some performance gains (Shao et al., 2024). We leave

*Table 8.* Exact pass@$k$ performance across models on Qwen2.5-7B-Instruct. This table contains the same data as Figure 2.

| Model | pass@1 | pass@2 | pass@4 | pass@8 | pass@16 | pass@32 | pass@64 | pass@128 |
|---|---|---|---|---|---|---|---|---|
| **IMO-AnswerBench** | | | | | | | | |
| Qwen2.5-7B-Instruct | 6.8 | 10.3 | 14.3 | 18.3 | 22.2 | 26.0 | 29.7 | 33.5 |
| GRPO | 7.7 | 11.2 | 15.0 | 18.6 | 22.3 | 25.9 | 29.6 | 32.8 |
| NuRL + GRPO | **8.0** | **11.4** | 14.8 | 18.2 | 21.3 | 24.4 | 27.6 | 30.8 |
| GRPO (Deepscaler) | 7.7 | 10.3 | 13.2 | 16.3 | 19.3 | 21.9 | 24.6 | 27.8 |
| DAIL (Ours) | 6.4 | 10.3 | **15.1** | **20.3** | **25.3** | **29.9** | **34.1** | **37.8** |
| **BeyondAIME** | | | | | | | | |
| Qwen2.5-7B-Instruct | 4.1 | 6.0 | 8.7 | 12.1 | 16.6 | 22.4 | 30.1 | 41.0 |
| GRPO | 4.2 | 6.4 | 9.4 | 13.2 | 18.0 | 23.9 | 31.0 | 40.0 |
| NuRL + GRPO | 3.3 | 5.4 | 8.0 | 11.1 | 15.1 | 20.2 | 27.0 | 36.0 |
| GRPO (Deepscaler) | **4.9** | **7.1** | 9.6 | 12.4 | 15.5 | 19.0 | 22.9 | 27.0 |
| DAIL (Ours) | 4.3 | 6.8 | **10.4** | **15.5** | **21.9** | **29.6** | **38.0** | **46.0** |
| **AIME 2024 / 2025** | | | | | | | | |
| Qwen2.5-7B-Instruct | 9.8 | 14.3 | 19.2 | 24.4 | 29.8 | 36.1 | 43.7 | 51.7 |
| GRPO | 9.0 | 13.2 | 17.6 | 22.6 | 29.1 | 37.1 | 45.4 | 53.3 |
| NuRL + GRPO | 11.0 | **16.3** | **21.5** | **26.4** | 31.5 | 36.8 | 42.0 | 46.7 |
| GRPO (Deepscaler) | **12.0** | 15.8 | 19.3 | 23.4 | 28.7 | 34.5 | 39.0 | 41.7 |
| DAIL (Ours) | 8.8 | 13.4 | 18.8 | 25.2 | **33.4** | **43.5** | **54.1** | **63.3** |

such an investigation for future work.

### C.8. Other Evaluation Details and Ablations

**Exact pass@$k$ results.** Exact numerical pass@$k$ results for Qwen2.5-7B-Instruct experiments can be found in Table 8.

**Naive baselines.** Figure 15 presents the version of Figure 3, i.e., the aggregated performance of training Qwen2.5-7B-Instruct on `e1-verifiable` across benchmarks, for all values of $k$.

**Effect of contrastive objective.** Table 9 presents the performance of various ablations of DAIL's contrastive objective on BeyondAIME when using mixed policy rollouts. We find that the contrastive objective outperforms optimizing only $D_{\mathrm{KL}}\big(M_\theta(\cdot|x, r_{<t}) \parallel M_{\mathrm{T}}(\cdot|r_{<t})\big)$. Additionally, minimizing the divergence between the student and the negative reference, i.e., optimizing $D_{\mathrm{KL}}\big(M_\theta(\cdot|x, r_{<t}) \parallel M_{\mathrm{NR}}(\cdot|r_{<t})\big)$ degrades performance compared to the untrained model at $k = 128$. These results match the motivation for this objective and negative reference outlined in §2.2.

**Comparing to standard distillation.** We also compare to standard supervised-distillation from Qwen3-235-A22B (Yang et al., 2025) using the same number of tokens per example as DAIL. The results in Table 10 demonstrate that both methods enable similar levels of coverage, making DAIL an attractive option as it does not require the use of a larger model for distillation.

**Competent models are necessary for consistent improvement with DAIL.** Figure 16 presents test-time scaling performance on smaller reasoning models, for which we find that the gains in performance are modest compared to the larger 8B and 14B models. Additionally, Table 11 presents BeyondAIME results on Llama-3.1-8B-Instruct, a much weaker model on mathematics tasks compared to the others used in our experiments. As shown, DAIL on `e1-verifiable` shows no improvement and in fact demonstrates a degradation in performance. This result demonstrates the limits of DAIL's ability to improve reasoning performance and the need for future work to understand the interaction between expert datasets and model strength.

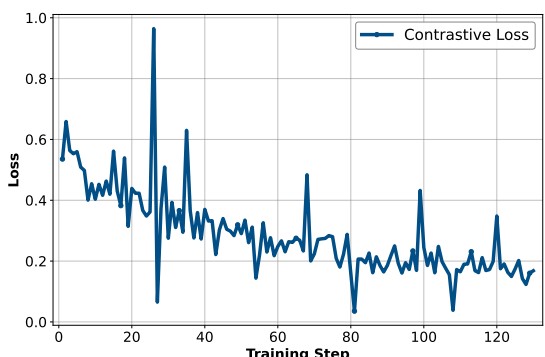

*Figure 13.* Loss curve for our contrastive loss. We find training is stable, with spikes in loss only occurring at epoch boundaries.

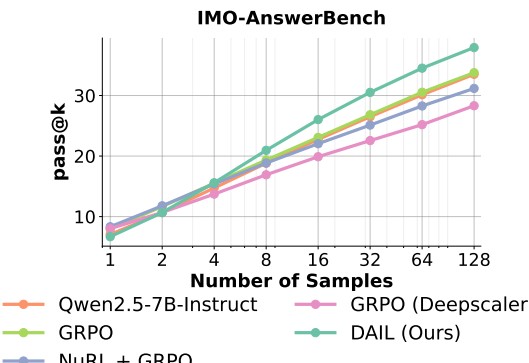

*Figure 14.* IMO-AnswerBench results with problems with the same source de-duplicated. There is little difference between these results and those in Figure 2.

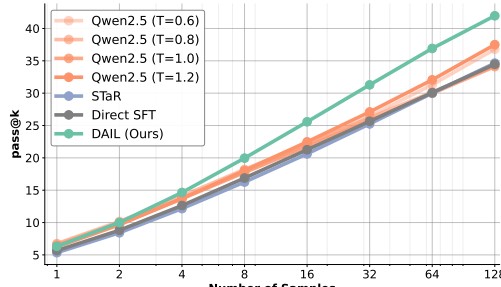

*Figure 15.* A version of Figure 3 for all $k$ values. These results include the aggregated performance of Qwen2.5-7B-Instruct when training on `e1-verifiable`.

*Table 9.* Ablation of contrastive objective on BeyondAIME. Results are in the mixed policy rollouts setting. Contrastive loss outperforms only minimizing KL between the student and teacher.

| Loss | pass@128 |
|---|---|
| Contrastive (Ours) | **46.0** |
| $D_{\mathrm{KL}}\big(M_\theta(\cdot\|x, r_{<t}) \,\|\, M_{\mathrm{T}}(\cdot\|r_{<t})\big)$ | 44.0 |
| Qwen2.5-7B-Instruct | 41.0 |
| $D_{\mathrm{KL}}\big(M_\theta(\cdot\|x, r_{<t}) \,\|\, M_{\mathrm{NR}}(\cdot\|r_{<t})\big)$ | 38.1 |

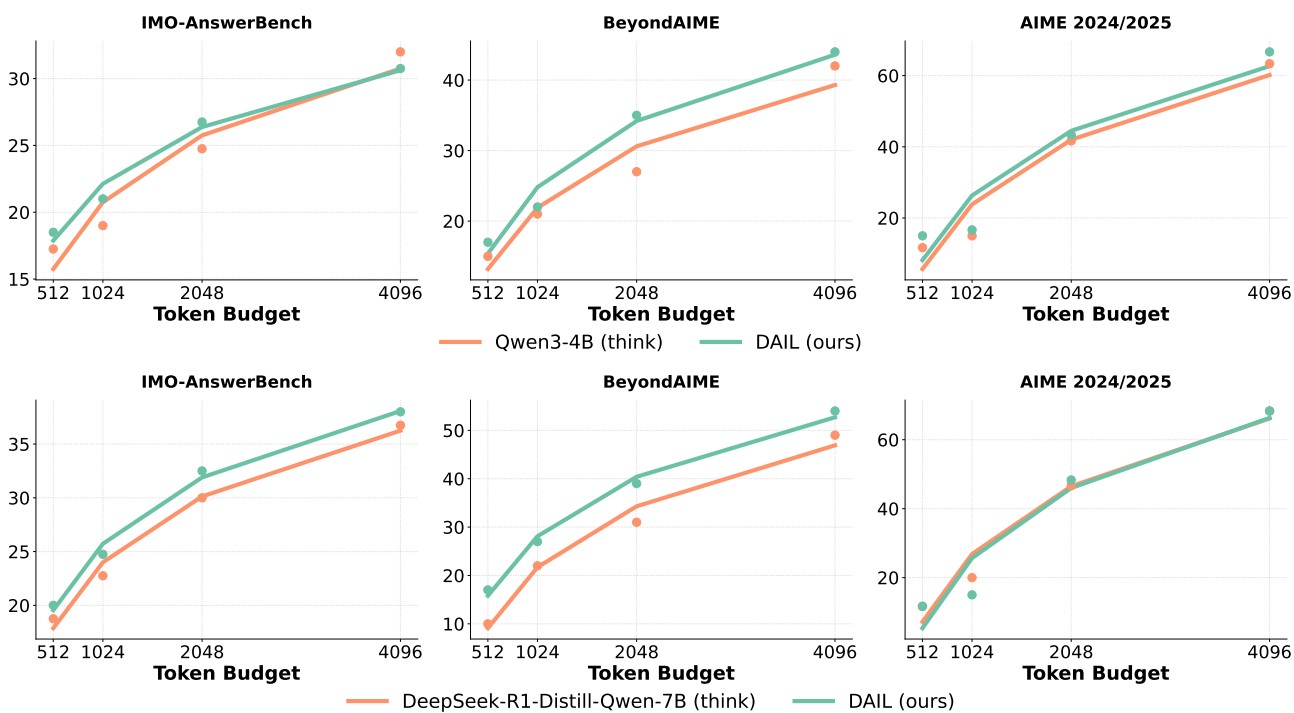

*Figure 16.* Test-time efficiency via coverage (pass@128) on smaller, less capable models. We find that for these less capable models, improvements are more modest, indicating that we need a baseline level of reasoning performance to apply DAIL for maximum improvement.

*Table 10.* Comparing DAIL to Distillation on Qwen (pass@128)

| Token Limit | DAIL | Distillation from Qwen3-235B-A22B (think) |
|---|---|---|
| 512 | **18.0** | 15.0 |
| 1024 | 23.0 | **24.0** |
| 2048 | 34.0 | **37.0** |
| 4096 | 49.0 | **55.0** |

**Performance on other backbones.**    Following Shao et al. (2025)'s recommendation to perform experiments on different backbones besides Qwen to understand whether a method generalizes beyond a specific model family, we also apply DAIL to a distilled Llama DeepSeek-R1 variant. The results in Table 12 demonstrate that DAIL consistently improves performance on the Llama-based model.

*Table 11.* Llama-3.1-8B-Instruct DAIL performance on BeyondAIME. Results demonstrate that weaker models may struggle to learn with DAIL if the difficulty of training problems is far outside their capacity.

| Model | pass@1 | pass@128 |
|---|---|---|
| Llama-3.1-8B-Instruct | **0.64** | 24.0 |
| DAIL | 0.47 | 24.0 |

*Table 12.* Performance of DeepSeek-R1-Distill-Llama-8B on BeyondAIME.

| Token Limit | DeepSeek-R1-Distill-Llama-8B | DAIL |
|---|---|---|
| 512 | 9.0 | **13.0** |
| 1024 | 15.0 | **29.0** |
| 2048 | 31.0 | **39.0** |
| 4096 | 45.0 | **57.0** |

