# OpenReview forum: "Making Expert Reasoning Learnable with Self-Distillation"
_ICML.cc/2026/Conference — ICML 2026 regular_

### Official Review · Reviewer_gHF5 · 2026-03-06

**Soundness:** 3
**Presentation:** 3
**Significance:** 4
**Originality:** 4
**Overall Recommendation:** 5
**Confidence:** 5

**Summary:**

This paper introduces DAIL, an imitation learning framework that allow LLM to learn from human expert solutions by transforming didactic expert traces into learnable, in-distribution reasoning paths. Experimental results demonstrate improvements in both task performance and reasoning efficiency.

**Compliance With Llm Reviewing Policy:**

Affirmed.

**Final Justification:**

Thank you for the detailed rebuttal. It addresses my main concerns, particularly about task diversity and methodological clarity, and it moderately improves my assessment.

Overall, I find the work sound and interesting, with reasonable originality and significance, and I have updated my recommendation accordingly.

**Key Questions For Authors:**

1. Since the experiments are mainly conducted on math tasks, how does the method compare with distillation from large reasoning models such as Qwen3-235B/DeepSeek-R1/GLM-5 when using the same amount of training data?
2. The second training stage uses a contrastive objective. How does it compare with preference-based methods such as DPO or KTO?
3. The approach appears to rely on the base model’s reasoning ability. How does it perform if the base model cannot generate meaningful reasoning traces initially?
4. For Qwen3-8B, only pass@1 results with a token budget are reported. What are the corresponding pass@k results?
5. The model is trained on a synthetic dataset that may include references to solution steps. What proportion of the data contains such references, and does the trained model tend to reproduce these patterns in its generated thinking tokens?

**Limitations:**

Yes

**Strengths And Weaknesses:**

**Strengths**

1. The paper addresses an important challenge: tasks where models fail to sample correct solutions, resulting in zero learning signals and making reinforcement learning ineffective.
2. The method demonstrates strong data efficiency, achieving improvements with fewer than 1000 samples.
3. The presentation is clear and logically organized.

**Weaknesses**

1. The evaluation shows limited task diversity, with both training and testing primarily focused on math tasks. Results on coding benchmarks such as LiveCodeBench would strengthen the study.
2. Exact numerical results are missing. Most results are presented only as line plots, making it difficult to distinguish performance when curves overlap at small sample sizes.
3. From Figure 2, DAIL improves pass@k mainly when (k) is large but appears to reduce pass@1, which is an important metric.

---

> ### Author Rebuttal · Authors · 2026-03-31
>
> Thank you for your detailed review! We are glad that you found that DAIL “addresses an important challenge” and “demonstrates strong data efficiency”. We have addressed your comments below:
>
>
> > The evaluation shows limited task diversity, with both training and testing primarily focused on math tasks.
>
>
> We applied DAIL to three new domains (health, reasoning puzzles, and tool calling):
>
>
> **Health Bench:**
> * Training: 1500 examples in the Consensus subset using the grading rubric and the expert response as the solution for the teacher.
> * Evaluation: 50 held-out examples from Hard subset (reporting total score and axis breakdown)
> |Model|Total Score|Accuracy|Communication|Completeness|Context Awareness|Instruction Following|
> |---|---|---|---|---|---|---|
> |Qwen2.5-7B-Instruct|0.16|0.31|0.59|0.21|0.00|**0.63**|
> |DAIL|0.16|**0.32**|**0.63**|**0.22**|**0.09**|0.57|
>
> **Reasoning Gym:**
> * Training: 500 maze task examples using the solution and minimum solution length as the solution for the teacher
> * Evaluation: 50 held-out examples
> |Model/k|1|2|4|8|16|32|64|128|
> |---|--|--|--|--|--|--|--|--|
> |Qwen2.5-7B|13.5|22.5|35.7|53.5|72.4|86.1|92.9|96.0|
> |DAIL|**20.6**|**34.3**|**52.2**|**71.1**|**85.9**|**94.8**|**99.2**|**100**|
>
> **Tool Use:**
> * Training: 4046 problems in the Tool Alpaca training split used by SDPO
> * Evaluation: 68 problem test set
> |Model/k|1|2|4|8|16|32|64|128|
> |---|--|--|--|--|---|---|---|----|
> |Qwen3-8B|57.7|61.0|64.0|66.8|69.1|71.1|73.1|75.0|
> |DAIL|**58.6**|**63.7**|**67.5**|**70.2**|**72.5**|**74.5**|**75.9**|**76.5**|
>
>
> > Most results are presented only as line plots
>
>
> We followed the convention of other papers, such as [1], to present pass@k results as a graph plotting k vs. performance. However, we will add tables with the exact numerical results to the Appendix.
>
>
> > From Figure 2, DAIL improves pass@k mainly when (k) is large but appears to reduce pass@1
>
>
> As we state in lines 255 - 257, for the difficult problems that we target, pass@1 is inherently low (≈ 5%), so the real utility of these models depends on their ability to uncover correct solutions via sampling (measured by pass@k, for high k). However, some models improve in both pass@1 and pass@k, such as Qwen3-14B and DeepSeek-R1-Distill-7B on BeyondAIME:
>
>
> |Tokens|DeepSeek-R1-Distill-7B Pass@1|DAIL (R1) Pass@1|Qwen14B Pass@1|DAIL (Qwen14B) Pass@1|
> |---|---|---|---|---|
> |512|1.8|**2.1**|2.6|**4.0**|
> |1024|2.8|**2.9**|3.2|**6.4**|
> |2048|6.1|**6.8**|4.5|**8.8**|
> |4096|12.9|**14.1**|7.0|**17.6**|
>
>
> > How does the method compare with distillation from large reasoning models?
>
> On  Qwen3-8B, we compare DAIL to distillation from Qwen3-235B, and find they perform similarly. We will add these results to the revised version:
>
>
> |Tokens|DAIL|Distillation|
> |---|---|---|
> |512|**51.0**|50.0|
> |1024|**60.0**|50.0|
> |2048|**62.0**|58.0|
> |4096|62.0|**66.0**|
>
>
>
>
> > How does contrastive objective compare to DPO or KTO?
>
>
> DPO and KTO operate at the sequence level. While effective for stylistic and safety alignment, they suffer from a sparse credit assignment problem in long-form reasoning, unable to pinpoint specific flawed steps. Conversely, DAIL operates at the token level, dynamically reweighting loss per position based on the divergence between the student, teacher, and negative reference. As detailed in Section 2.2, this fine-grained control explicitly penalizes "rationalization shortcuts" (reaching a correct conclusion via flawed or missing logic), which sequence-level methods like DPO cannot distinguish from valid reasoning.
>
>
>
>
> > Effect on performance if the base model cannot generate meaningful reasoning traces initially?
>
>
> Since the teacher receives the ground truth human solution in its context, it does not need to be able to solve the problem to generate a correct solution, but rather simply needs to comprehend and fill in didactic gaps. As we note in the introduction, this approach alleviates the zero-reward issue with standard outcome-based RL, where the base model fails to generate a correct reasoning trace.
>
>
>
>
> > What are the corresponding pass@k results for Qwen3-8B?
>
>
> Figure 4 actually presents the pass@128 results across token budgets. We will make this clear by updating the caption and Section 3.3.
>
>
> > What proportion of the data contains references to solution steps, and does the trained model tend to reproduce these patterns?
>
>
> While 7.9% of the Qwen2.5-7B-Instruct training data cites a "reference solution," only 3.1% of the trained model's evaluated rollouts reproduce this. Crucially, rollouts with these citations are 13% less likely to be correct than the sample average. Manual inspection confirms these rare cases are not systematic overfitting, but a failure mode where the struggling model hallucinates an authoritative source to rationalize flawed logic. This demonstrates that DAIL internalizes the privileged student's core reasoning while actively suppressing unhelpful stylistic artifacts.
>
>
> [1] https://arxiv.org/pdf/2504.13837

---

> > ### Author Rebuttal · Reviewer_gHF5 · 2026-04-03
> >
> > Thank the authors for their detailed rebuttal. Based on their response, I have updated my recommendation accordingly.

---

### Official Review · Reviewer_xcXX · 2026-03-12

**Soundness:** 3
**Presentation:** 2
**Significance:** 3
**Originality:** 2
**Overall Recommendation:** 4
**Confidence:** 4

**Summary:**

This paper introduces DAIL (Distribution Aligned Imitation Learning), a novel approach to enhance the reasoning capabilities of Large Language Models by effectively leveraging high-quality expert solutions. The authors identify a fundamental "didactic gap" where expert-written proofs, intended for human readers, contain implicit reasoning jumps and lack the internal cognitive dynamics of a model, leading to poor performance when used for naive imitation learning. To bridge this gap, DAIL employs a privileged-student distillation process that transforms didactic expert solutions into constructive, model-aligned reasoning paths through mixed policy rollouts. Furthermore, the method incorporates a contrastive loss objective to penalize "rationalization shortcuts", thereby enforcing rigorous step-by-step deduction. Experimental results on Qwen-based models demonstrate that DAIL achieves accuracy gains and token efficiency on challenging benchmarks.

**Compliance With Llm Reviewing Policy:**

Affirmed.

**Final Justification:**

Thank you for the author's reply, which has resolved my doubts. Therefore, I will maintain the "weak accept" opinion but correspondingly increase my confidence.

**Key Questions For Authors:**

1. How does DAIL perform on non-Qwen architectures and non-mathematical domains?
2. DAIL relies on "privileged students" to generate model-aligned reasoning. If this student model generates a path that reaches the correct answer through flawed or hallucinated logic, how does the framework prevent the target model from learning these "distinguished hallucinations"?

**Limitations:**

No, see weakness and questions.

**Strengths And Weaknesses:**

**Strength**
1. This method can achieve a 10-25% performance gain using fewer than 1000 high-quality expert solutions, and has high sample efficiency.
2. This method can learn directly from expert proofs, alleviating the training difficulties of RL in fields where automatic scoring is not possible.

**Weakness**
1. The experiments only test mathematical reasoning and Qwen-based model.
2. If there are hidden logical errors in the alignment paths generated by "privileged students", the model will also absorb them as correct knowledge.

---

> ### Author Rebuttal · Authors · 2026-03-31
>
> Thank you for your thoughtful review! We are glad that you found that DAIL has “high-sample efficiency” while “alleviating the training difficulties of RL”. We have addressed your comments below:
>
>
> > How does DAIL perform on non-Qwen architectures and non-mathematical domains?
>
>
> We agree that it is valuable to evaluate how DAIL generalizes across different architectures and domains.
>
>
> First, to address architecture generalization, we applied DAIL to DeepSeek-R1-Distill-Llama-8B (a Llama-based model, rather than Qwen). As shown below, DAIL yields significant pass@128 improvements on BeyondAIME across all token budgets, demonstrating that our method is architecture-agnostic:
>
>
> |Tokens Limits|DeepSeek-R1-Distill-Llama-8B|DAIL|
> |-------------|-----------------------------|----|
> |512|9.0|**13.0**|
> |1024|15.0|**29.0**|
> |2048|31.0|**39.0**|
> |4096|45.0|**57.0**|
>
>
>
>
> Additionally, we evaluated DAIL on three new domains to show it can generalize well beyond mathematics tasks:
>
> **Health Bench:**
> * Training: 1500 examples in the HealthBench Consensus dataset using the grading rubric and the ideal expert response as the solution for the teacher.
> * Evaluation: 50 held-out examples from HealthBench Hard (reporting total score and axis breakdown)
> |Model|Total Score|Accuracy|Communication|Completeness|Context Awareness|Instruction Following|
> |---|---|---|---|---|---|---|
> |Qwen2.5-7B-Instruct|0.16|0.31|0.59|0.21|0.00|**0.63**|
> |DAIL|0.16|**0.32**|**0.63**|**0.22**|**0.09**|0.57|
>
> **Reasoning Gym:**
> * Training: 500 maze task examples using the solution and minimum solution length as the solution for the teacher.
> * Evaluation: 50 held-out examples
> |Model/k|1|2|4|8|16|32|64|128|
> |---|--|--|--|--|--|--|--|--|
> |Qwen2.5-7B|13.5|22.5|35.7|53.5|72.4|86.1|92.9|96.0|
> |DAIL|**20.6**|**34.3**|**52.2**|**71.1**|**85.9**|**94.8**|**99.2**|**100**|
>
> **Tool Use:**
> * Training: 4046 problems in the Tool Alpaca training split used by SDPO
> * Evaluation: 68 problem test set
> |Model/k|1|2|4|8|16|32|64|128|
> |---|--|--|--|--|---|---|---|----|
> |Qwen3-8B|57.7|61.0|64.0|66.8|69.1|71.1|73.1|75.0|
> |DAIL|**58.6**|**63.7**|**67.5**|**70.2**|**72.5**|**74.5**|**75.9**|**76.5**|
>
>
>
>
> > DAIL relies on "privileged students" to generate model-aligned reasoning. If this student model generates a path that reaches the correct answer through flawed or hallucinated logic, how does the framework prevent the target model from learning these "distinguished hallucinations"?
>
>
> As we note in Section 2.2, the privileged student’s expanded reasoning traces may contain rationalization shortcuts, which are deficient logical bridges to an intermediate result contained in the expert solution - the flawed logic you described. Later, starting on line 140, we explain how we create the negative reference model, which is conditioned on waypoints in the solution, shifting the model’s distribution to favor forcing or jumping directly to intermediate results. This behavior characterizes rationalization shortcuts, so to penalize it, we add the KL divergence between the negative reference and the student (the model being trained) as a negative weighted term in the loss.

---

> > ### Author Rebuttal · Reviewer_xcXX · 2026-04-04
> >
> > Thanks for the rebuttal. I will keep my score.

---

### Official Review · Reviewer_BZJE · 2026-03-13

**Soundness:** 3
**Presentation:** 3
**Significance:** 4
**Originality:** 4
**Overall Recommendation:** 4
**Confidence:** 4

**Summary:**

This work talks about Distribution Aligned Imitation Learning (DAIL), a method to improve increase reasoning capability of LRMs on hard problems by resolving existing issues in RL training and imitation learning. RL training suffers from 0-gradient on hard problems and imitation learning suffers from human solutions being out-of-distribution (OOD). This gap is bridged by Generating In-Distribution Traces and Contrastive Learning to Mitigate Shortcuts.

**Compliance With Llm Reviewing Policy:**

Affirmed.

**Final Justification:**

The authors have address all my new concerns thoroughly. I will keep my score of 4 but I have increased my confidence.

**Key Questions For Authors:**

1. The method introduces some hyperparameters like confidence threshold and the divergence weight. What is the sensitivity of these hyperparameters. It has not been discussed.
2. I want to know if after the applying this method, how does the performance on simple questions change. Does it degrade, improve or remain the same.
3. What is the exact computational and latency overhead of executing Mixed Policy Rollouts compared to standard direct sampling?
4. If the base model (e.g., a 1.5B or 3B parameter model) is too weak, does the privileged student fail to generate mathematically sound bridges for the didactic shortcuts, thereby poisoning the synthetic dataset?

**Limitations:**

1. The method is tested only math tasks. Other domains tasks testing would have made a more significant impact.

**Strengths And Weaknesses:**

Strengths.
1. The problem is veery well formulated and is extremenly timely.
2. Strong Empirical Validation
3. Highly sample efficient.

For weakness check below.

---

> ### Author Rebuttal · Authors · 2026-03-31
>
> Thank you for your thoughtful review! We are glad that you found the paper “well formulated” and “extremely timely”. We have addressed your comments below:
>
> > The method introduces some hyperparameters like the confidence threshold and the divergence weight. What is the sensitivity of these hyperparameters?
>
> During the rebuttal period, we ran a sweep over the confidence threshold ($\tau \in$ {0.6, 0.7, 0.8, 0.9}) and found only small deviations in Qwen3-8B pass@128 performance on BeyondAIME with a 2048 token limit:
>
> |$\tau$|DAIL|
> |------|--------|
> |0.6|61.0|
> |0.7|60.0|
> |0.8|62.0|
> |0.9|59.0|
>
> As we note in Appendix B.3, $\gamma$, the divergence weight hyperparameter, was tuned on the 100-question validation set over the following values {0.01, 0.05, 0.1, 0.2, 0.3, 0.4, 0.5, 0.6, 0.7, 0.8, 0.9, 1.0}. We find that training is stable for $\gamma \le 0.2$, but outside of this range, the negative reference exerts too much negative pressure on the policy, leading to instability.
>
> We will add this analysis to the Appendix in the camera-ready version of the paper.
>
> > After applying this method, how does the performance on simple questions change? Does it degrade, improve, or remain the same?
>
> We agree that it is important to understand how DAIL impacts performance on easier in-domain mathematics questions and general reasoning questions. We performed an analysis on 500 questions from the MMLU Pro No Math, which assesses general knowledge and reasoning ability, and MATH-500, a mathematics reasoning dataset, which is easier than the three explored
>
> **MMLU Pro No Math:**
>
> |Model/k|1|2|4|8|16|32|64|128|
> |---|---|---|---|---|---|---|---|---|
> |Qwen2.5-7B|52.6|55.7|58.3|60.5|62.3|63.6|64.4|65.0|
> |DAIL|**53.2**|**57.1**|**60.3**|**62.9**|**64.9**|**66.4**|**67.3**|**67.8**|
>
> **MATH 500:**
>
> |Model/k|1|2|4|8|16|32|64|128|
> |---|---|---|---|---|---|---|---|---|
> |Qwen2.5-7B|**71.6**|**78.1**|**82.6**|85.8|88.1|89.9|91.3|93.0|
> |DAIL|69.3|77.0|82.4|**86.3**|**89.2**|**91.3**|**92.7**|**93.4**|
>
>
> Based on these results, it's clear that DAIL performs similarly to the base model on both out-of-distribution questions and easier in-distribution questions. We will add this result to the appendix of the revised version of the paper.
>
> > What is the overhead of executing Mixed Policy Rollouts compared to standard direct sampling?
>
> Empirically, our current implementation has a latency of approximately 3x compared to standard direct sampling using vLLM. To cleanly intervene in the decoding process, our implementation utilizes two vLLM engines on separate GPUs and a token-wise generation loop in which we pull logits to the CPU to apply our custom probability-based routing at every step. We temporarily bypass vLLM's internal continuous batching, which incurs a scheduling overhead per token.
>
> We believe it is possible to achieve ~ 1x latency, but this would require incorporating our sampling logic directly into the vLLM engine. However, because DAIL uses Mixed Policy Rollouts strictly as an **offline data generation mechanism** to create synthetic reasoning traces, and not as a real-time inference strategy, this latency increase is heavily mitigated by the massive parallelization of the dataset generation process, as noted in line 180.
>
>
> > If the base model is too weak, does the privileged student fail to generate mathematically sound bridges for the didactic shortcuts, thereby poisoning the synthetic dataset?
>
> We experimented with running DAIL with the e1-verifiable dataset on Llama-3.1-8B-Instruct, which is weaker at mathematical reasoning compared to Qwen2.5-7B-Instruct:
>
> |Model|pass@1|pass@128|
> |---|---|---|
> |Llama-3.1-8B-Instruct|**0.64**|24.0|
> |DAIL|0.47|24.0|
> This result indicates that DAIL cannot be used to teach models with a dataset of arbitrarily hard problems, as learning seems to become ineffective if there is too much of a gap between the model capabilities and the difficulty of the dataset. We believe this is an important piece of analysis for practitioners using this method, and will add this discussion to the camera-ready version of the paper.
>
> > Testing on other domains would have made a more significant impact.
>
> We applied DAIL to three new domains (health, reasoning puzzles, and tool calling):
>
>
> **Health Bench:**
> |Model|Total Score|Accuracy|Communication|Completeness|Context Awareness|Instruction Following|
> |---|---|---|---|---|---|---|
> |Qwen2.5-7B-Instruct|0.16|0.31|0.59|0.21|0.00|**0.63**|
> |DAIL|0.16|**0.32**|**0.63**|**0.22**|**0.09**|0.57|
>
> **Reasoning Gym:**
> |Model/k|1|2|4|8|16|32|64|128|
> |---|--|--|--|--|--|--|--|--|
> |Qwen2.5-7B|13.5|22.5|35.7|53.5|72.4|86.1|92.9|96.0|
> |DAIL|**20.6**|**34.3**|**52.2**|**71.1**|**85.9**|**94.8**|**99.2**|**100**|
>
> **Tool Use:**
> |Model/k|1|2|4|8|16|32|64|128|
> |---|--|--|--|--|---|---|---|----|
> |Qwen3-8B|57.7|61.0|64.0|66.8|69.1|71.1|73.1|75.0|
> |DAIL|**58.6**|**63.7**|**67.5**|**70.2**|**72.5**|**74.5**|**75.9**|**76.5**|

---

> > ### Author Rebuttal · Reviewer_BZJE · 2026-04-03
> >
> > Thanks for the rebuttal. I already gave a score of 4 (weak accept).

---

### Official Review · Reviewer_d9Da · 2026-03-14

**Soundness:** 2
**Presentation:** 3
**Significance:** 4
**Originality:** 4
**Overall Recommendation:** 5
**Confidence:** 5

**Summary:**

The paper falls under a class of parallely emerging LLM RL methods that convert reference solutions into more on-policy reasoning traces. Specifically, the proposed method does the following. First, the student is conditioned on the expert solution to produce a reasoning trajectory. However, for reasoning models the trajectory can contain unnatural references to the solution, and less self-correction steps than one might face on a novel problem at inference time. To fix this, the method only replaces student tokens by the privileged student's tokens when the privileged student assigns the student generated token probability below a fixed threshold (hyperparameter). The paper calls this mixed policy rollouts. Further, to mitigate expert solution conditioned rollouts skipping important steps, negative reference rollouts are constructed for contrastive learning which artificially skip key steps. Crucially, this entire process can be done off-policy based on the initial model's rollouts which allows for greater efficiency. The method is used to train on two math datasets collected by the authors (417 AIME problems from 1985 to 2023, and 669 proof-based math olympiad problems) which are also released. Upon training, the hyperparameter tuned proposed method outperforms baselines with untuned default hyperparameters in pass@k, and token efficiency. The paper concludes with interesting ablations showing how their contrastive objective (albeit with extra hyperparameter tuning) improves over NLL, some interesting signs of generalization, and the mixed policy rollouts also help.

**Compliance With Llm Reviewing Policy:**

Affirmed.

**Final Justification:**

I think the paper should be accepted, as the method does improve pass@k at high k=4 onwards compared to the status quo, GRPO, with evidence from math benchmarks in the paper, and more diverse datasets in the rebuttal. The paper also does a great job of working through the details of on-policy imitation methods and is educational to read. However, the method also has worse at pass@1 and pass@2 than GRPO which can be considered a limitation. Future work can explore converting the improved pass@k obtained from this method into higher pass@1, though had this paper achieved this I would have recommended a full 6 rating.

**Key Questions For Authors:**

The abstract has the line "expert solutions are typically didactic, containing implicit reasoning gaps intended for human readers rather than computational models". I wonder whether a solution intended for human readers is fundamentally different than one that a language model can understand. This is an interesting hypothesis, but would need to be tested not assumed. Language models, being trained to imitate human text in large parts, are in some way a simulation of humans. It is thus not obvious to me that the ideal solutions for them to learn from would look different. Do the authors have any evidence to support this central assumption?

A small suggestion. Some of the phrases in the paper like "didactic shortcuts" vs "rationalization shortcuts" sound like jargon which is not very common. For purely readability improvements, it would be nice if the authors could add clear definitions or examples of this somewhere early in the paper. I know Figure 1 tries to do this but I found it hard to follow.

Why have a fixed uniform threshold on the token probabilities (btw its not clearly defined M is the softmaxed probability, I inferred it from the range [0, 1]) for deciding whether to keep the student generated token or privileged student? The token probabilities are often highly contextual, and using a fixed threshold for this seems suboptimal. One natural alternative is having a learnt policy for deciding whether to keep the student or teacher token, though its unclear what the best objective for this "router" policy would be when trying to train the student (maybe training it on maximizing rollout accuracy). An important lower overhead baseline is parallel "self-distillation" work like https://arxiv.org/abs/2601.20802. Could you please compare with this as I'm sure readers would be interested in knowing which method works best? FYI, since this is parallel work, I will not downgrade scores even if your method performs worse. In fact, I am likely to increase my support even if that is the case, if you present this result transparently!

Can we also maybe call the privileged student the "teacher" throughout to avoid confusion, while clarifying early on that the teacher is actually just the same policy but with privileged information?

I appreciate that the method can work with off-policy rollouts. But I would still like to see whether this sacrifices performance compared to doing the same process a bit more on-policy, for e.g. by generating rollouts with the same process at intermediate stages of optimization. Do you have any experiments on this?

**Limitations:**

I think the impact statement is well written, though the paper does not acknowledge limitations.

**Strengths And Weaknesses:**

### Strengths

- I think the paper does a pretty good job of not just proposing a privileged information based on-policy training method, but also reasoning through potential things that could go wrong and trying to fix them.

- I like the insight on how direct sampling is enough for Instruct models, but not for reasoning models!

- I think the datasets curated and released in this work could actually be one of the most impactful contributions of it, even though this isnt highlighted a lot! Clearly the focus was on the method, which makes me even more appreciative that the authors went out of their way to collect 417 AIME problems instead of the usual small sets, and also collected math olympiad proof problems.

- I really like the experimental section of the paper, and overall presentation.

### Weaknesses

- I feel like a bunch of assumptions are made about why simpler methods would fail in Section 2. The paper proposes some intuitions for this, but I would much rather like to see direct references to experiments that test them out. I see there are some experiments that somewhat map to these assumptions between Figure 3,5,6. Could you map these more explicitly for readers so its also easier to evaluate the quality of evidence for each precise hypothesis?

- As mentioned in Appendix B.4, there is no hyperparameter tuning on the baselines like GRPO and NuRL. This is problematic for obvious reasons, as the hyperparameters for the proposed method were tuned. I understand this is expensive, but still emphasise the authors should do this so we can get an accurate picture of how much the proposed method actually improves over the default. I think this will go a long way in convincing the community about the proposed method.

- This is opinionated, but I think the negative reference contrastive learning is inelegant. It feels like a waste of compute to produce rationalized shortcut-laden reasoning traces. I would much prefer a simpler method that works without this step. How confident are the authors that such a step is essential to the on-policy self-distillation paradigm this paper falls under?

- Most of the experiments are on math datasets, except GPQA-Diamond, but that also very often has numerical questions. I think the method is righly pitched to be general, so I would love to see whether it works well for e.g. on non-verifiable (e.g. HealthBench), long-horizon (e.g. Reasoning Gym) and Tool-Use benchmarks (e.g. search tasks). Proof of success here would definitely shift my score up, in fact, if you show all three (negative results are okay), along with answering my remaining questions below to a reasonable extent, I will be happy to go up to 6, and champion the paper for recognition at the venue :)

Minor: The abstract mentions 10-25% pass@k gains, but does not mention a) which dataset this is on, and b) what the value of k is, which makes it impossible to interpret these numbers at this stage.

---

> ### Author Rebuttal · Authors · 2026-03-31
>
> Thank you for the detailed review! We are glad that you found the paper’s analysis, insights, and contribution of new datasets valuable to the community. We have addressed your comments below:
>
> > Add direct references from the assumptions about simpler methods to results
>
> Thank you for the suggestion. In the camera-ready version, we will explicitly map our assumptions to the empirical results showing DAIL's superiority over simple baselines (Fig. 3), the benefits of our contrastive loss over NLL (Fig. 5), and DAIL's resistance to overfitting compared to baselines (Fig. 6).
>
> > Hyperparameter tuning on RL baselines
>
> During the rebuttal period, we swept over lr $\in$ {1e-6, 5e-6, 1e-5} and temperature $\in$ {0.6, 1.0} for the GRPO baseline. After tuning, the baseline does exhibit improved performance, but still underperforms the instruct model (37% pass@128 for tuned GRPO vs. 41% for Qwen2.5 base, compared to 46% for DAIL on BeyondAIME). We will update Figure 2 with these results.
> > Is the negative reference term in the contrastive loss necessary?
>
> To clarify, tokens are not generated using the negative reference. Instead, it defines a distribution used to compute the KL divergence with the student, as shown in the loss equation (line 160). Thus, DAIL only requires *forward passes* through the negative reference during training. We will make this computational advantage (briefly noted in line 186) more explicit. Furthermore, our loss ablation (Fig. 15) shows that while the full contrastive objective is optimal, optimizing only the teacher-student KL still outperforms the baseline. So, the negative reference is beneficial but is not strictly required.
> > How does DAIL perform on non-verifiable, long-horizon, and tool-use benchmarks?
>
> We have evaluated DAIL on these three sets of tasks. The results are provided below:
>
> **Health Bench:**
> * Training: 1500 examples in the Consensus subset using the grading rubric and the expert response as the solution for the teacher.
> * Evaluation: 50 held-out examples from Hard subset (reporting total score and axis breakdown)
> |Model|Total Score|Accuracy|Communication|Completeness|Context Awareness|Instruction Following|
> |---|---|---|---|---|---|---|
> |Qwen2.5-7B-Instruct|0.16|0.31|0.59|0.21|0.00|**0.63**|
> |DAIL|0.16|**0.32**|**0.63**|**0.22**|**0.09**|0.57|
>
> **Reasoning Gym:**
> * Training: 500 maze task examples using the solution and minimum solution length as the solution for the teacher
> * Evaluation: 50 held-out examples
> |Model/k|1|2|4|8|16|32|64|128|
> |---|--|--|--|--|--|--|--|--|
> |Qwen2.5-7B|13.5|22.5|35.7|53.5|72.4|86.1|92.9|96.0|
> |DAIL|**20.6**|**34.3**|**52.2**|**71.1**|**85.9**|**94.8**|**99.2**|**100**|
>
> **Tool Use:**
> * Training: 4046 problems in the Tool Alpaca training split used by SDPO
> * Evaluation: 68 problem test set
> |Model/k|1|2|4|8|16|32|64|128|
> |---|--|--|--|--|---|---|---|----|
> |Qwen3-8B|57.7|61.0|64.0|66.8|69.1|71.1|73.1|75.0|
> |DAIL|**58.6**|**63.7**|**67.5**|**70.2**|**72.5**|**74.5**|**75.9**|**76.5**|
>
> > Comparison to parallel SDPO work
>
> DAIL’s pass@1 (58.6) on Tool Alpaca is less than the 68.5 reported by SDPO. We attribute this to two factors: SDPO’s wall-clock evaluation complicates direct sample-efficiency comparisons, and Tool Alpaca provides sparse feedback (only final tool calls). While DAIL excels at learning from human expert traces,  SDPO seems better suited for environments constrained to sparse, outcome-level signals.
>
> > Is a solution intended for human readers different than one that a language model can understand?
>
> While Figures 3 and 5 implicitly address this, we agree that an explicit comparison strengthens our claim: optimizing NLL on teacher-sampled solutions rather than raw human-written ones yields a 21.1% pass@128 improvement (Qwen2.5-7B-Instruct, across all three math benchmarks). This empirically confirms that LLMs learn significantly better from our synthetic traces than from raw human solutions.
>
> >Some phrases in the paper could be more clearly defined
>
> We will add clearer definitions for didactic and rationalization shortcuts in the introduction and Figure 1.
>
> >  Fixed uniform threshold on the token probabilities.
>
> Yes, we compare the model’s softmaxed probability against the threshold to decide whether to sample from the privileged student. We will clarify this in the revised version. Learning a policy to decide whether to keep the student’s token is an interesting idea that we hope future work explores.
>
> > Change privileged student to the "teacher" throughout to avoid confusion
>
> Yes, we will update the terminology in the updated version.
>
> > How well does DAIL work with on-policy rollouts?
>
> Early experiments with on-policy training showed extremely slow loss convergence, requiring many epochs on our intentionally small dataset. Combined with a >10x increase in step time due to on-policy sampling overhead, training to convergence was computationally impractical in our setup. We will discuss these trade-offs in the appendix.

---

> > ### Author Rebuttal · Reviewer_d9Da · 2026-04-03
> >
> > Thanks for the new results and clarifications. Could you also report GRPO results for the three new datasets to put the DAIL results into perspective?

---

> > > ### Author Response · Authors · 2026-04-06
> > >
> > > Thank you for your response. Below, we present the GRPO results on Reasoning Gym and Tool Alpaca. Since HealthBench is non-verifiable, it is difficult to train with RL on this task, which is a motivator for DAIL, as we mention in line 78 in the manuscript.
> > >
> > > **Reasoning Gym**
> > >
> > > *Maze Task:*
> > >
> > > |Model/k|1|2|4|8|16|32|64|128|
> > > |---|--|--|--|--|--|--|--|--|
> > > |Qwen2.5-7B|13.5|22.5|35.7|53.5|72.4|86.1|92.9|96.0|
> > > |DAIL|20.6|34.3|**52.2**|**71.1**|**85.9**|**94.8**|**99.2**|**100**|
> > > |GRPO|**43.5**|**45.8**|47.3|47.9|48.0|48.0|48.0|48.0|
> > >
> > > *Rotten Oranges:* We also report results on the harder Rotten Oranges task in the Reasoning Gym environment:
> > > |Model/k|1|2|4|8|16|32|64|128|
> > > |---|--|--|--|--|--|--|--|--|
> > > |Qwen2.5-7B|5.0|9.1|15.9|25.9|39.2|54.3|67.2|74.0|
> > > |DAIL|10.5|**18.9**|**31.4**|**47.1**|**62.7**|**74.8**|**83.6**|**92.0**|
> > > |GRPO|**14.1**|17.3|20.8|25.6|32.4|40.9|51.1|64.0|
> > >
> > >
> > > **Tool Use (Tool Alpaca)**
> > > |Model/k|1|2|4|8|16|32|64|128|
> > > |---|--|--|--|--|--|--|--|--|
> > > |Qwen3-8B|57.7|61.0|64.0|66.8|69.1|71.1|73.1|75.0|
> > > |DAIL|58.6|63.7|**67.5**|**70.2**|**72.5**|**74.5**|**75.9**|**76.5**|
> > > |GRPO|**64.5**|**65.5**|66.3|66.8|67.2|67.5|67.6|67.6|
> > >
> > >
> > > **Analysis:** Unlike the main tasks in the paper, which contained problems that were explicitly chosen to be either unsolvable (pass@$k$ = 0 for high $k$) or non-verifiable, these two domains have higher pass rates, making them more amenable to improvement via online RL. While these datasets are outside the difficult-problem setting that we considered in the paper, there are valuable insights from these results. Specifically, GRPO only exhibits improvements at lower $k$ and degrades pass@k performance at $k \ge 8$ relative to the untrained model, likely due to a severe loss of generation diversity (mode collapse). On the other hand, DAIL yields consistent performance improvements compared to the base model **across $k$-values**. We believe this analysis is important to help practitioners understand when and how to use DAIL, and we will add these results and a discussion of this point to the camera-ready version of the paper.

---

### Decision · Program_Chairs · 2026-04-30

**Decision:**

Accept (regular)

**Comment:**

The paper proposes DAIL, which parses expert trajectories through the student model so that demonstrations become in-distribution for the student and easier to learn from. DAIL shows strong empirical performance with fewer expert demonstrations, outperforming multiple baselines and improving reasoning efficiency. The paper also releases the resulting in-distribution reasoning dataset.

All reviewers agreed on the significance of the method and the value of the curated dataset. The rebuttal further demonstrated the generalization of the proposed approach across architectures, domains, and hyperparameter settings. We therefore recommend acceptance, while encouraging the authors to better position their work relative to recent papers in this space (e.g., approaches that leverage the model's own generation conditioned on expert trajectories or answers to ensure the training data is in-distribution), such as https://arxiv.org/abs/2601.18779, https://arxiv.org/pdf/2601.20802, https://arxiv.org/pdf/2507.02834, etc.